# PHIPS-HALO: the airborne particle habit imaging and polar scattering probe - Part 3: Single Particle Phase Discrimination and Particle Size Distribution based on Angular Scattering Function

Fritz Waitz[1], Martin Schnaiter[1,2], Thomas Leisner[1], and Emma Järvinen[1,3]

[1]Institute of Meteorology and Climate Research, Karlsruhe Institute of Technology, Karlsruhe, Germany
[2]SchnaiTEC GmbH, Karlsruhe, Germany
[3]National Center for Atmospheric Research (NCAR), Boulder, CO, USA

**Correspondence:** F. Waitz (fritz.waitz@kit.edu) and E. Järvinen (emma.jaervinen@kit.edu)

**Abstract.**

A major challenge for in-situ observations in mixed phase clouds remains the phase discrimination and sizing of cloud hydrometeors. In this work, we present a new method to determine the phase of individual cloud hydrometeors based on their angular light scattering behaviour employed by the PHIPS airborne cloud probe. The phase discrimination algorithm is based on the difference of distinct features in the angular scattering function of spherical and aspherical particles. The algorithm is calibrated and evaluated using a large data set gathered during two in-situ aircraft campaigns in the Arctic and Southern Ocean. Comparison of the algorithm with manually classified particles showed that we can confidently discriminate between spherical and aspherical particles with a 98% accuracy. Furthermore, we present a method to derive particle size distributions based on single particle angular scattering data for particles in a size range from $100\,\mu m \leq D \leq 700\,\mu m$ and $20\,\mu m \leq D \leq 700\,\mu m$ for droplets and ice particles, respectively. The functionality of these methods is demonstrated in three representative case studies.

## 1 Introduction

Mixed-phase clouds, consisting of both supercooled liquid droplets and ice particles, play a major role in the life cycle of clouds and the radiative balance of the earth (e.g. Korolev et al. (2017)). Despite their widespread occurrence, mixed-phase cloud processes are still rather poorly understood and represent a great source of uncertainty for climate predictions (e.g. McCoy et al. (2016)). As a consequence, more in-situ observations are needed to better understand mixed-phase cloud processes and improve climate models. Microphysical properties and life cycle of mixed-phase clouds are strongly dependent on the phase separation of liquid and ice phase (e.g. Korolev et al. (2017)). Furthermore, the radiative properties of cloud particles depend on their phase, shape and size. Despite the importance of mixed-phase cloud phase composition, a major uncertainty remains in the correct phase discrimination of cloud hydrometeors.

Currently, phase discrimination of individual cloud particles larger than 200 $\mu$m is based on circularity analysis (e.g. diameter- or area ratio, Cober et al. (2001)) of ice particle images measured by optical array probes such as the 2DS and 2DC (*Two-dimensional Stereo Probe*, *Two-dimensional Cloud Probe*, SPEC Inc., Boulder, USA) or CIP (*Cloud Imaging Probe*, DMT, Longmont, USA). For smaller particles, such discrimination methods of optical array probes are limited due to their optical resolution, especially for out of focus particles (Korolev (2007)). Instruments utilizing optical microscopy, such as the *Cloud Particle Imager* (CPI, SPECinc, Boulder, USA), have a finer resolution and are able to discriminate particles down to 35 $\mu$m McFarquhar et al. (2013). Still, the phase discrimination between droplets and quasi-spherical or small irregular ice particles based on their images can be challenging, as shown in Fig. 1.

For very small particles below $D \leq 50 \mu$m, the SID-family of instruments like the *Small Ice Detector Mark 3* (SID-3, Vochezer et al. (2016)) and *Particle Phase Discriminator* (PPD, Hirst and Kaye (1996); Kaye et al. (2008); Vochezer et al. (2016),; Mahrt et al. (2019)) offer reliable phase discrimination based on the spatial distribution of the forward scattered light. The SID family of instruments has the disadvantage, however, that they do not measure the phase of each single particle but only for a sub-sample. Therefore, a large sampling time is required to derive ice concentrations in mixed-phase clouds that are dominated by droplets. The *Cloud and Aerosol Spectrometer with Polarization* (CAS-POL, DMT, Longmont, USA, Glen and Brooks (2013)) is an instrument that measures the light scattered by single cloud particles and aerosols in a size range of $0.6 \mu m \leq D \leq 50 \mu$m in the forward and backward directions. Based on the polarization ratio of the backscattered light, the sphericity of the cloud particles can be determined (Sassen (1991); Nichman et al. (2016)). However, recent studies have suggested, that particle phase discrimination of polarization-based measurements can misclassify up to 80% of the ice particles as droplets in the presence of small, quasi-spherical ice (Järvinen et al. (2016)).

Hence, in the size range $D \leq 100 \mu$m, methods for reliable particle phase discrimination, are still needed. The *Particle Habit Imaging and Polar Scattering* probe (PHIPS) is a unique instrument designed to investigate the microphysical and light scattering properties of cloud particles. It produces microscopic stereo-images whilst simultaneously measuring the corresponding angular scattering function from $18°$ to $170°$ for single particles in a size range from $50 \mu m \leq D \leq 700 \mu$m and $20 \mu m \leq D \leq 700 \mu$m for droplets and ice particles, respectively. More information and a detailed characterization of the PHIPS setup and instrument properties can be found in depth in Abdelmonem et al. (2016) and Schnaiter et al. (2018).

In this work, we will present a method to discriminate the phase of single cloud particles based on their angular scattering function. An algorithm was developed using experimental in-situ data from two aircraft campaigns targeting mixed-phase clouds. We present a method to use single-particle angular light scattering measurements to produce size distributions for spherical and aspherical particles separately.

This work is structured in the following: in section 2, the aircraft campaigns during which the experimental data sets used in this work, are introduced. Next, in section 3 the methodology and calibration of the phase discrimination algorithm are explained. In section 4, the particle sizing will be introduced and several methods for shattering correction will be discussed. Finally, in section 5, the described methods will be used in three case studies. The results will be compared to measurements by other cloud particle probes during the same campaigns.

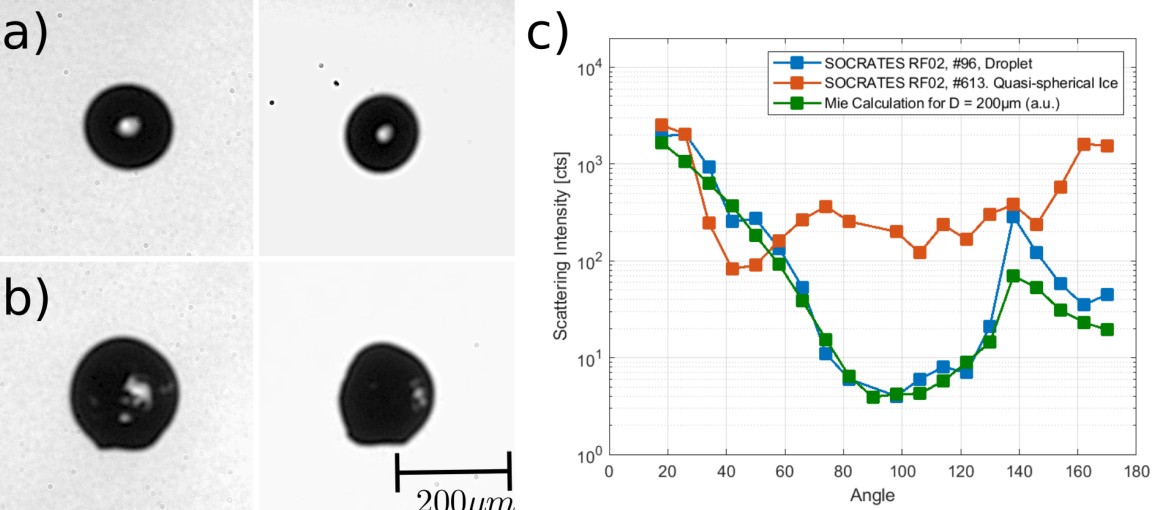

**Figure 1.** Stereo micrograph of a droplet (a) and a quasi-spherical ice particle (b) taken by the PHIPS probe. In the stereo micrograph, the two views of the particle have an angular distance of 120°. The instrument concurrently recorded the angular light scattering functions of the imaged particles as displayed in (c). The theoretical scattering function calculated for a droplet with a diameter of 200 μm calculated using the Mie theory is shown for comparison in (c). The calculated scattering-intensity is integrated over the field of view of each of PHIPS' 20 polar nephelometer channels so it can be compared to the measurement (see supplementary information S6 for details).

## 2   Experimental Data Sets

In this work, we use experimental in-situ data gathered during two airborne field campaigns to develop and test a single-particle phase discrimination algorithm for the PHIPS probe. The two data sets refer to the two respective campaigns:

1. ACLOUD - Arctic CLoud Observations Using airborne measurements during polar Day, May/June 2017 based in Svalbard (Spitsbergen, Norway) and

2. SOCRATES - Southern Ocean Clouds, Radiation, Aerosol Transport Experimental Study, Jan/Feb 2018 based in Hobart (Tasmania, Australia).

An overview of the meteorological and microphysical conditions as well as the instrumentation during those campaigns can be found in Knudsen et al. (2018) and Wendisch et al. (2019) for ACLOUD and McFarquhar et al. (2019) for SOCRATES. The sampling during both campaigns includes a wide variety of different cloud conditions: warm clouds, supercooled liquid clouds, ice clouds and mixed-phase clouds. Clouds were sampled in an altitude range from boundary layer clouds below 200 m to mid-level clouds between 4000 m and 6000 m. Temperatures ranged from -15 to +5 °C during ACLOUD and -35 to +5 °C during SOCRATES. The sampled ice particles covered a range of different particle shapes and habits (columns, plates, needles, bullet rosettes, dendrites and irregulars, including rough, rimed and pristine particles) as well as sizes. More details can be found in the supplementary material (S1). The instrumentation on the two aircraft included cloud particles probes such as the SID-3,

CDP (*Cloud Droplet Probe*, DMT, Longmont, USA), CIP and PIP (*Precipitation Imaging Probe*, DMT, Longmont, USA) during ACLOUD and 2DC, 2DS and CDP during SOCRATES. Due to the variability of the meteorological conditions and sampled particles, the data gathered during these two campaigns makes a suitable and representative data set to develop the phase discrimination and particle size distribution algorithms that are presented in this work.

## 3   Single-Particle Phase Discrimination Algorithm

The angular scattering properties of spherical particles can be analytically calculated using Mie theory. The angular scattering properties of usually aspherical ice particles, however, are much more complex, which significantly alters their scattering properties compared to spherical particles (Järvinen et al. (2018); Schnaiter et al. (2018); Sun and Shine (1994); Um and McFarquhar (2011)). Hence, it is possible to differentiate between the angular scattering functions (ASF) of spherical and aspherical particles by looking into differences in the angular light scattering behaviour in the angular regions where spherical

particles exhibit unique features, like the minimum around 90° and the rainbow around 140°. In this section, we introduce four scattering features and develop an algorithm that is able to classify each particle based on the combined information from multiple features of the ASF (see Fig. 2).

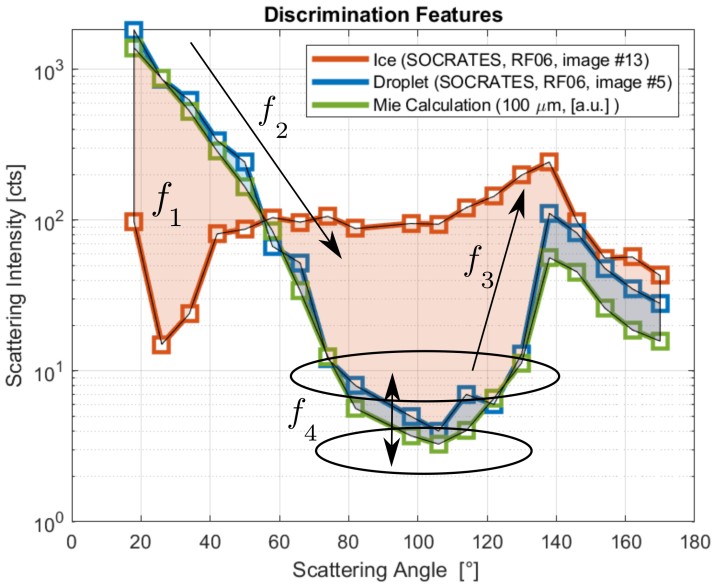

**Figure 2.** Visualization of the four classification features: $f_1$ = Mie comparison (shaded area between curves and Mie calculation), $f_2$ = down slope, $f_3$ = up slope before the rainbow feature and $f_4$ = ratio around the minimum at 90°. The green line shows the calculated ASF for a theoretical spherical particle. the blue and lines show the measured ASF of an exemplary droplet (D = 119.6 µm) and ice crystal (D = 165.8 µm) from the SOCRATES campaign.

The basic concept of the development procedure for the single-particle phase discrimination algorithm will be explained in this section and is shown in Fig. 3. In the first step, ASFs calculated by Mie theory (BHMIE, Bohren and Huffmann) for spherical particles using the refractive index for water ($n_{refr}$ = 1.332) are compared to modelled ASFs of aspherical ice crystals (Baum et al. (2011) and Yang et al. (2013)). Based on the differences in the ASFs, typical features are determined that are characteristic for spherical or aspherical particles (see Fig. 2). The algorithm is then calibrated and validated using PHIPS data from the two field campaigns that were introduced in the previous section. This data set consists of about 23,000 representative single cloud particles of various phase, habit and size for which stereo micrographs as well as the corresponding ASFs are available. Those particles are manually classified as spherical or aspherical based on their appearance in the stereo micrographs. The calibration of the phase discrimination algorithm is then based on the ACLOUD data set only. This way, a classification probability for every feature is determined. The different features are then weighed and combined to a final discrimination probability for every single particle. Lastly, the data from the SOCRATES campaign is used to validate the discrimination algorithm and to determine the discrimination accuracy.

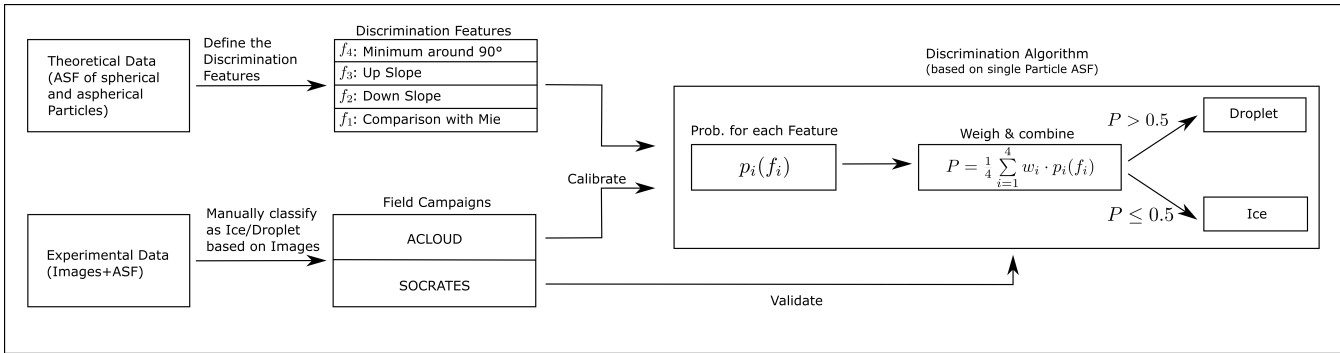

**Figure 3.** Schematics showing the basic working principle of the phase discrimination algorithm.

## 3.1 Discrimination Features

### 3.1.1 $f_1$: Comparison with Mie Scattering

One approach to discriminate between spherical and aspherical particles is to compare a particle's ASF with theoretical Mie calculation. To estimate the deviation of the observed ASF from the calculated Mie scattering, we evaluate the integrated difference between measurement and calculation (shaded area between the curves in Fig. 2). Fig. 4 shows a step-by-step explanation of the determination of the $f_1$ parameter based on two exemplary droplets: a droplet (d1) with D = 119.6 µm (the same particle as used in Fig. 2) and a theoretical Mie-sphere (d2) with d = 200 µm. Fig. 4a shows the ASF for the two particles as well as the ASF of the reference Mie-sphere with D = 100 µm.

We define the ratio between the measured intensity $I_{\text{exp}}$ of an individual particle and the Mie calculation $I_{\text{Mie}}$ for a spherical reference particle with a diameter of 100 µm for every nephelometer angle $\theta_i$

$$q(\theta_i) = \frac{I_{\text{exp}}(\theta_i)}{I_{\text{Mie}}(\theta_i)} \tag{1}$$

as shown in Fig. 4b. To be comparable to the measured intensities, the calculated theoretical Mie scattering function was integrated over the field of view of the polar nephelometer channels (see Supp. S6). Ideally, this ratio $q_i$ should be calculated

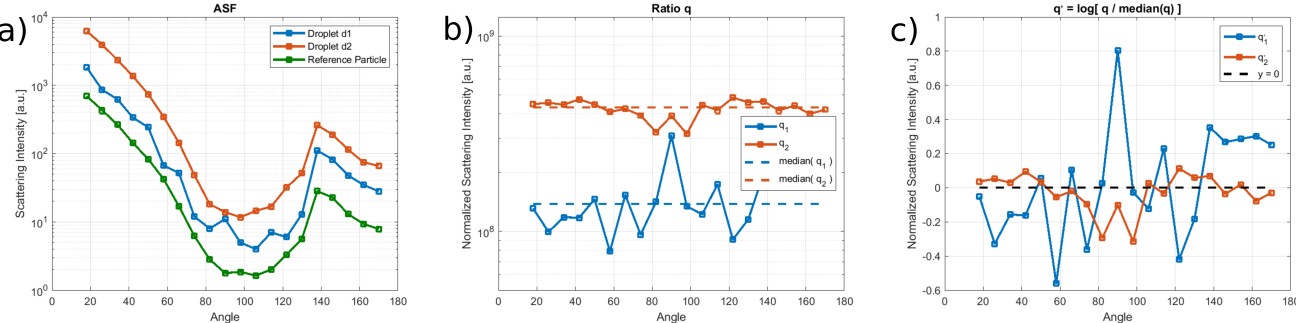

**Figure 4.** Determination of the feature parameter $f_1$ of two exemplary droplets: Droplet d1 (blue) is the same particle as used in Fig.2. Droplet d2 (red) is a theoretical Mie-sphere with d = 200 µm. The plots show the particles' ASF (a), $q$ and $\bar{q}$ (b) and $q'$ (c). The resulting $f_1$ is then calculated as the integral over all channels (i.e. area between each curve and y = 0). The resulting values are $f_1 = 3.7$ for d1 and $f_1 = 2.2$ for d2.

with a theoretical reference particle with the same diameter as the detected particle. However, the diameter of the measured particle is not known without applying a size calibration first. To circumvent this, each $q_i$ is normalized by the median over all channels $\bar{q}$ (dashed line in Fig. 4b). For a spherical particle, this ratio should be approximately $q \simeq const.$ (see Supp. S2). Since we do not know the diameter of the measured particle without applying a size calibration, $q$ is normalized by the median over all channels $\bar{q}$ and the influence of the approximately constant factor can be neglected. This also has the advantage, that we do not need to calibrate the conversion factor from counts to power unit $(W)$ of the photomultiplier array which can change for different campaigns, gain settings and changes in laser power. Thus, the discrimination algorithm works for different campaigns and settings without further calibration.

Furthermore, as the deviation in 'both directions' from the calculated Mie intensity have to be weighted equally, i.e. $q_i = 2$ and $q_i = \frac{1}{2}$ should be equivalent. Therefore, we make the transformation $q'_i \to \log(q_i/\bar{q})$. The resulting 'feature parameter' is then finally defined as the logarithm of the integral over all angles $\theta_i$:

$$f_1 = log\left(\int |q'_i| \, d\theta_i\right) = log\left(\int \left|\log\left(\frac{q_i}{\bar{q}}\right)\right| d\theta_i\right) \tag{2}$$

which corresponds to the area under the curves in Fig. 4c.

To demonstrate that this feature is representing a distinctive difference between spherical and aspherical particles, the distribution of the feature parameter value $f_1$ of representative, manually classified spherical and aspherical particles from the

experimental in-situ aircraft measurement campaigns introduced in section 3.3, are shown in Fig. 6a. It can be seen that, roughly, if a given particle has a feature value of e.g. $f_1 < 4.5$, it is likely spherical, if $f_1 > 5$, it has a high probability of being an aspherical particle. Phase discrimination based on this feature alone would already allow a reasonable discrimination, but there also exist spherical particles with e.g. $f_1 > 5$ that would be misclassified by using this approach. Hence, multiple features are taken into account to increase the discrimination accuracy.

### 3.1.2 $f_2 + f_3$: Down and Up-Slope

When looking at Fig. 2, the most distinctive differences between the ASF of spherical and aspherical particles are the minimum around $90°$ and the rainbow maximum around $140°$ for spherical particles, whereas aspherical particles often show a flatter angular scattering behaviour. One way to extract those features is to evaluate the 'exponential slope'

$$f_2 = \frac{\log(I(\theta_2)) - \log(I(\theta_1))}{\theta_2 - \theta_1} \tag{3}$$

in the region before and after the minimum around $90°$. This results in two features: the negative slope before the minimum and the positive slope between minimum and rainbow around $140°$. In general, steeper slopes mean that a given particle is likely to be spherical. The first 'slope feature' ($f_2$) is the 'Down Slope', which is simply the linear slope from $\theta_1 = 42°$ to $\theta_2 = 74°$. The first three scattering channels ($\Theta = 18°, 26°, 34°$) are not taken into account here, because they have a larger possibility to be saturated for larger particles. The slopes are determined by applying a linear fit to the logarithmic intensities in the channels between $\theta_1$ and $\theta_2$.

The second slope feature ($f_3$), the 'Up Slope', is calculated as the (logarithmic) slope from the minimum around $90°$ to the maximum of the rainbow peak. Since the scattering intensity can be very low and, therefore, comparable to the magnitude of the background noise (especially for small particles), hence the 'lower end' is averaged over multiple channels from $\theta = 74°$ to $106°$. The upper end of the slope is not fixed either, but rather chosen dynamically as the angular position of the rainbow peak can vary within four scattering channels between $\theta = 130°$ and $154°$. Thus, we define the slope feature $f_3$ as

$$f_3 = \frac{\log\left(\max[I(130° \text{ to } 154°)]\right) - \log\left(\text{mean}[I(74° \text{ to } 106°)]\right)}{\theta_2 - \theta_1}, \tag{4}$$

with the corresponding angle of the rainbow maximum $\theta_2$ and the minimum $\theta_1 = 90°$. This way, even small particles and elongated particles with a shifted rainbow peak (see App. **??**) can be classified correctly.

### 3.1.3 $f_4$: Ratio around the 90° Minimum

Another possible way to depict the depth of the $90°$ minimum is to directly compare the intensities in the vicinity around $\theta = 90°$ with channels that are farther away (see Fig. 2). Hence, the 'Mid Ratio' feature is defined as

$$f_4 = \log\left(\frac{\text{mean}[I(58°, 66°, 114°, 122°)]}{\text{mean}[I(74°, 82°, 90°, 98°, 106°)]}\right). \tag{5}$$

With the distinct shape of the ASF of droplets around the $90°$ minimum one could argue that an intensity threshold might be enough to discriminate between spherical and aspherical particles (e.g. classifying every particle with $I(\theta = 90°)$ smaller than

a certain threshold $I_{\text{thresh}}$ as spherical). However, looking at absolute values would prove impractical as the ASF scales with particle size: a very small aspherical particle could still fulfil $I(\theta = 90°) < I_{\text{thresh}}$ as well as a rather large spherical particle

$I(\theta = 90°) > I_{\text{thresh}}$, respectively. Hence, the discrimination features presented here are all based on relative values, slopes and ratios instead of discrete thresholds. Further, all discrimination features are based on the scattering signal of multiple channels instead of only one channel to minimize the impact of noise. This allows the discrimination algorithm to be used for multiple campaigns (even with differing settings or minor hardware changes or malfunction) without additional calibration (see section 3.4).

## 3.2    Simulation of the Feature Parameters

To prove that the defined set of discrimination features reliably discriminates between spherical and aspherical particles, we calculate the feature parameter values $f_i$ based on theoretical ASF. For droplets, we use Mie theory for spherical particles with diameters from $100\,\mu\text{m} \leq \text{D} \leq 700\,\mu\text{m}$. For ice, we use modelled orientation-averaged ASF of ice crystals of different habits and roughness using the databases from Baum et al. (2011) and Yang et al. (2013) in the size range from $20\,\mu\text{m} \leq \text{D} \leq 700\,\mu\text{m}$.

Similarly as explained beforehand, the scattering intensities are integrated over the field of view of the polar nephelometer channels. The distribution of feature parameters is shown in Fig. 5. It can be seen, that the resulting values differ significantly for droplets and ice. This shows, that the aforementioned features are in fact fit to discriminate the ASF of spherical and aspherical particles. From now on, we will assume that particles that appear spherical in terms of their angular light scattering behavior are droplets and particles that appear aspherical in their ASF are ice. Note that this includes also deformed droplets

(as discussed in the supplementary material S4) as well as quasi-spherical ice as shown in Fig. 1.

### 3.3    Calibration

Next, the discrimination features were applied to experimental data sets of real cloud particles. We used in-situ data of representative, manually classified single particles to validate the calculated features. This experimental data was then used to calibrate the algorithm (i.e. the classification probability functions $P_i(f_i)$ for every feature), in order to have a numerical function that

calculates a classification probability for every feature of a given particle, and later a combined probability that can be used to discriminate every single particle based on its phase.

     The experimental data sets used for the calibration and verification of the discrimination algorithm are described in detail in section 2. As it is the goal to develop an algorithm that is suitable without any further calibration for upcoming campaigns, the calibration and verification data sets are entirely disjunct: the ACLOUD data set is used for calibration, the verification is done

using the SOCRATES data set. The ACLOUD and SOCRATES campaigns comprise 14 and 15 research flights, during which, in total about 41,000 and 235,000 single particles were detected by PHIPS, respectively. More details about sizes and habits of the manually classified particles used for the calibration can be found in the supplementary material (S1). Because the imaging component of PHIPS has a limited temporal resolution, this results in about 22,000 and 32,000 events with matching stereo micrographs for the ACLOUD and SOCRATES flights, respectively. Based on these stereo micrographs, all imaged particles

were manually classified as ice or droplets. To ensure a representative data set, only clearly distinguishable particles were

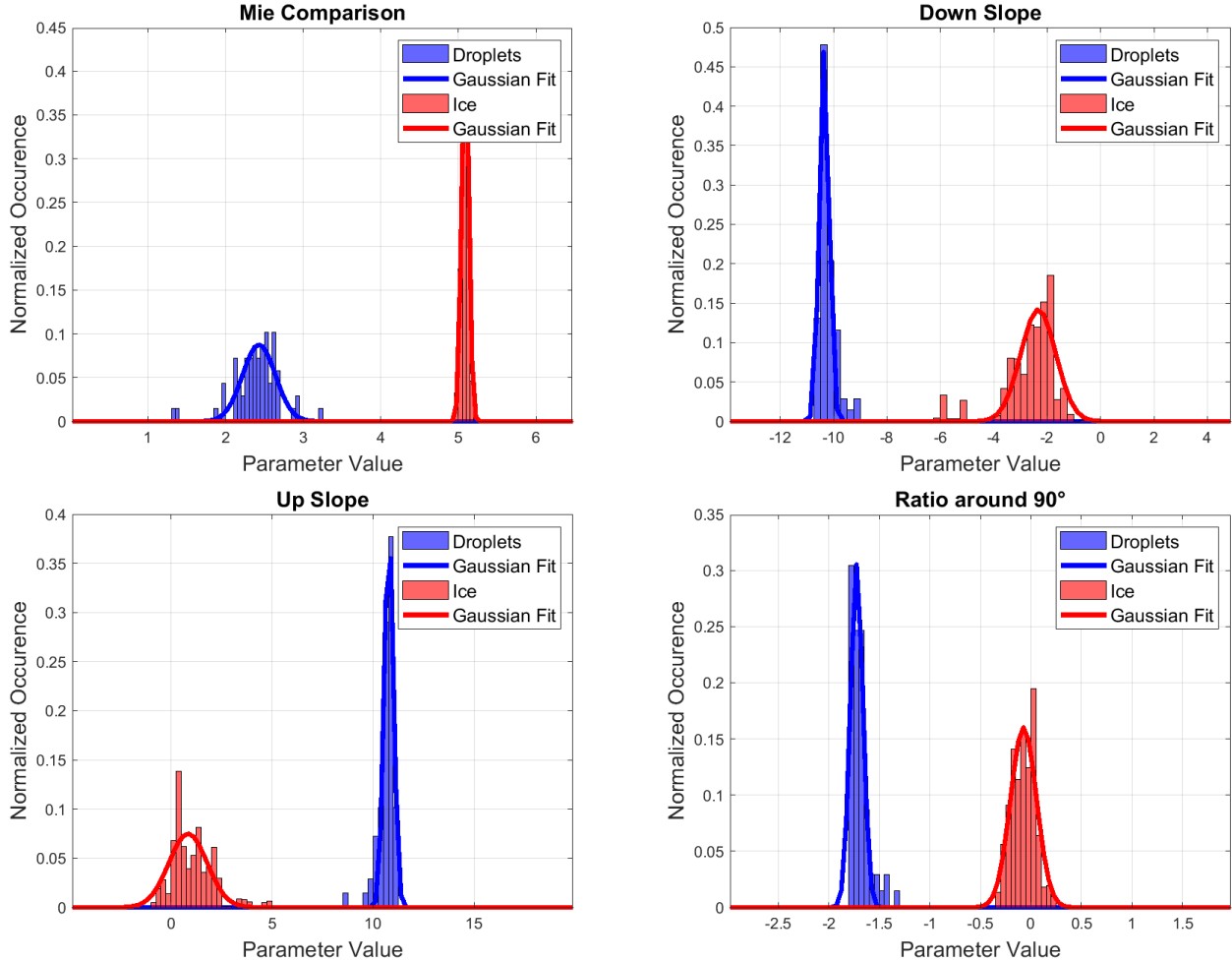

**Figure 5.** Normalized histograms of the discrimination features, $f_i$, evaluated for theoretical ASFs. Simulated ASFs were calculated using Mie theory in case of droplets (blue) and by selecting typical ice particle habits (red) from the light scattering databases by Baum et al. (2011) and Yang et al. (2013). Normal distribution fits to the data are depicted by solid lines in the graphs. Note that the simulations provide orientation-averaged ASF whereas the observed particles by PHIPS have random but fixed orientation.

taken into account, whereas images that show multiple particles and particles that are only partly imaged, out of focus or not clearly distinguishable, were ignored. Hence, the resulting data set used for the calibration (based on the ACLOUD campaign) includes 1,853 droplets and 7,885 ice crystals. The data set used for the validation and determination of the discrimination accuracy (see section 3.4) contains of 2,284 droplets and 9,936 ice crystals from the SOCRATES campaign. The chosen data sets consist of representative cloud particles which cover a wide range of different particle shapes and habits (columns, plates, needles, bullet rosettes, dendrites and irregulars, including rough, rimed and pristine particles) as well as sizes D = 20 - 700 μm and D = 100 - 700 μm for ice and droplets, respectively.

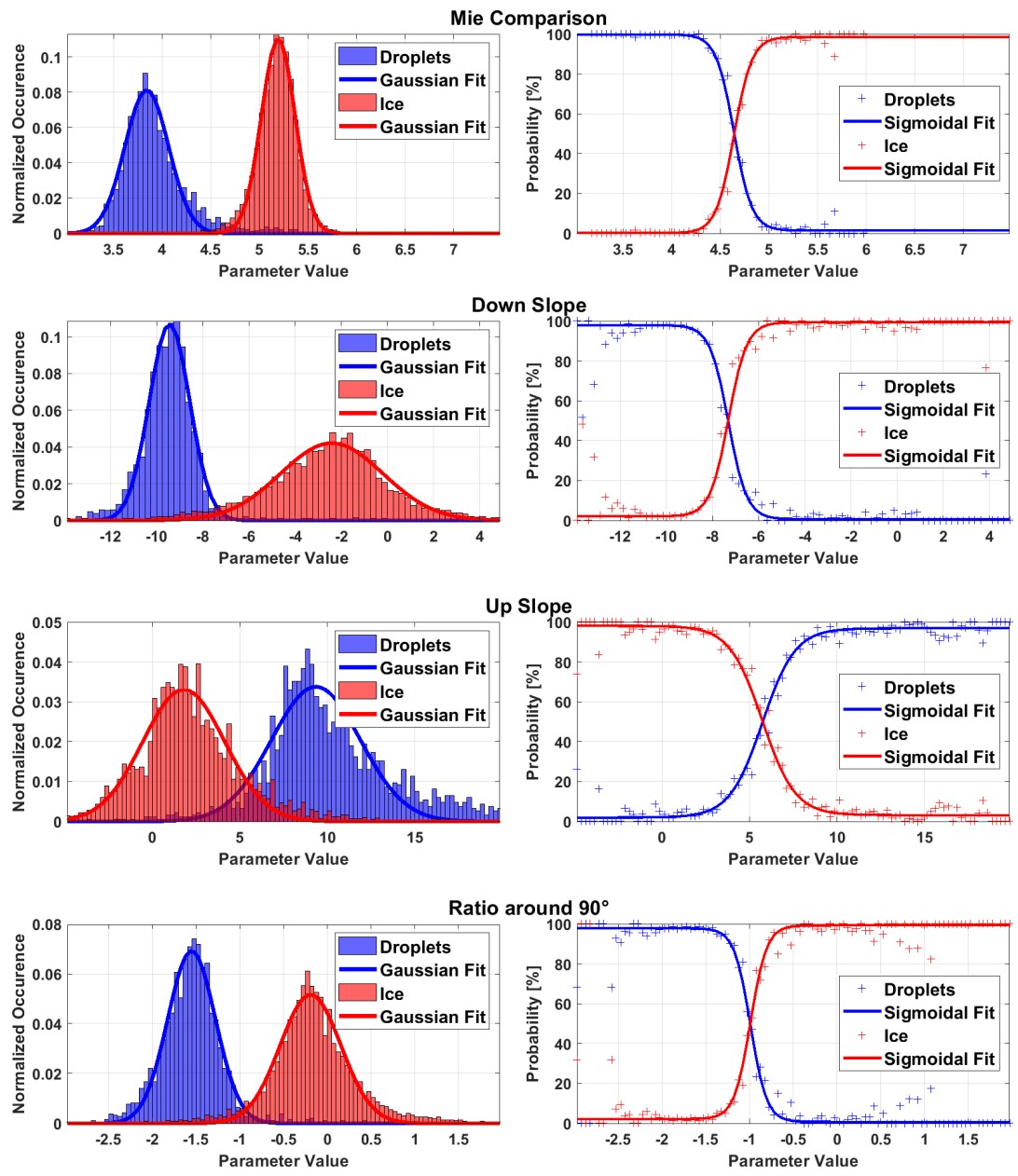

**Figure 6.** Left: normalized histograms of the discrimination features, $f_1$, $f_2$, $f_3$ and $f_4$, of all manually classified particles (blue: droplets, red: ice) from the ACLOUD campaign that were used for the calibration of the discrimination algorithm. The histograms can be nicely fitted by normal distributions (solid lines). Right: corresponding probability for a given particle with a given feature parameter value to be classified as ice or droplet, including sigmoidal fits.

The left panels of Fig. 6 show, similar to the simulations in Fig. 5, the relative amount $n(f_i)$ of particles that share a certain feature parameter value $X$. To account for the different amount of ice and droplets in the data set ($N_{\text{ice}} \approx 3 \cdot N_{\text{droplet}}$), the number frequencies $n_{\text{droplet/ice}}$ are normalized by the total amount of droplets and ice particles. The plots show that the distribution of the four aforementioned feature parameters are clearly distinct for droplets and ice and thus represent features that can be used to discriminate droplets from ice. Further, it can be seen that these normalized occurrences $n(f_i)$ are normally distributed. The distributions of the four feature parameters based on the measurements (Fig. 6) show a similar trend to the simulations (Fig. 5). The width of the distributions of feature parameters for measurements is much broader compared to the simulations. This can be explained by the single-orientation of the measured crystals compared to the orientation-averaging that was used in the simulations. Orientation-averaging tends to smooth out features in the ASFs and thus cause more narrow feature parameters. It should be also noted that the theoretical computations are for idealised crystals. Nevertheless, the mean values of the distributions agree very well. The only exception to this is the mean value of the distribution of droplets for $f_1$, which is shifted slightly to larger values compared to the simulations. This is to be expected because the "Mie-comparison-feature" $f_1$ is based on the relative difference between the measured and calculated ASF. This difference is much smaller for simulated particles as discussed in 3.1.1.

However, Fig. 6 also shows that the ice and droplets modes are not always clearly separable for every feature and for every particle. Therefore, instead of using a sharp threshold, a classification probability

$$P_i(f_i) = \frac{n_{\text{ice}}(f_i)}{n_{\text{ice}}(f_i) + n_{\text{droplet}}(f_i)}, \tag{6}$$

that a particle is classified as ice (or with $1 - P_i(f_i)$ as a droplet) based on the ratio between $n_{\text{droplet}}(f_i)$ and $n_{\text{ice}}(f_i)$ for each feature (see right panels of Fig. 6), is defined. Assuming that the $n_i(f_i)$ follow normal distributions with comparable widths, $P_i(f_i)$ can be approximated and fitted by a sigmoid function. Following that, the probability functions $P_i(f_i)$ are determined by using a sigmoidal fit for every feature based on the empiric data. These probabilities, $P_i$, for each feature are combined to

$$P_{\text{combined}} = \frac{1}{n} \sum_{i=1}^{n} w_i \cdot P_i(f_i) \tag{7}$$

with empiric weights $w_i$ that are determined using recursive, linear optimization. Coincidentally, the optimum weight is to weigh all four features equally, i.e. $w_1 = w_2 = w_3 = w_4 = 1$ and thus $P_{\text{combined}} = \text{mean}(P_i)$. Finally, this results in a classification probability for every given particle with a set of calculated feature parameter values $\{f_1, f_2, f_3, f_4\}$, which is then classified based on $P_{\text{combined}}$ as a droplet ($P \leq 50\%$) or ice particle ($P > 50\%$). Details on the fit parameters for $P_i$ can be found in App. A and B.

## 3.4 Discrimination Accuracy

Discrimination algorithms often run in danger of "overtraining" or creating a "lookup table", resulting in seemingly very good discrimination accuracies that, in reality, are just recreating the "training data" used for calibrating the system but fail to classify new, "unknown" data sets. In order to avoid this, the "training" and "test" data set are not only disjunct, but from

entirely different field campaigns. Furthermore, this proves that the algorithm is able to function independently for different

campaigns without further calibration.

The confusion matrices (Fawcett (2006)) for the discrimination algorithm for the two campaigns is shown in Fig. 7. For the SOCRATES data set, 99.7% of ice particles could be correctly classified as ice and only 29 out of 9,936 were misclassified as droplets. 95.8% droplets were classified correctly and 95 out of 2,284 were misclassified as ice. In total, out of all particles, 99.0% were classified correctly. Respectively, if a particle is classified as ice (droplet) by the algorithm, the expected error (i.e.

the probability that the initial particle was actually a droplet) amounts to 0.9% (1.3%). Also, 100% of the theoretical particles used in section 3.2 (which were not used for the calibration) were classified correctly. More details about the discrimination accuracy and misclassified particles can be found in the SI.

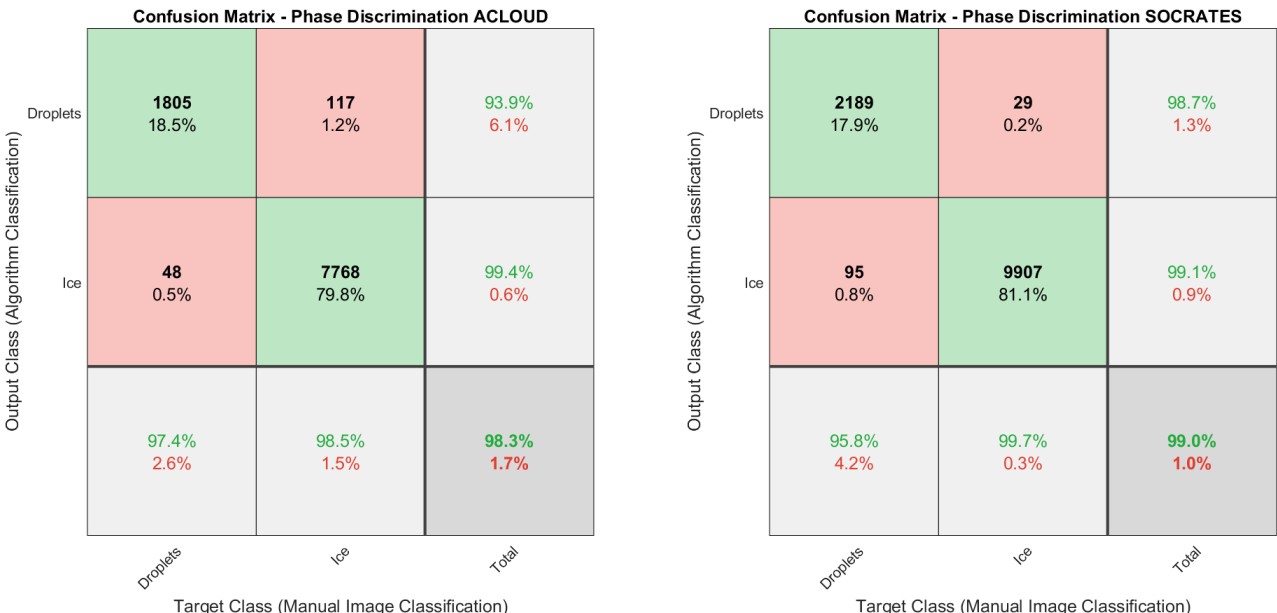

**Figure 7.** Confusion matrices that visualize the classification accuracy of the ice discrimination algorithm. The discrimination algorithm was applied to all manually classified particles from both the ACLOUD (left) and SOCRATES (right) data sets. In both cases the combined probability $P_{combined}$ from the ACLOUD calibration was used to calculate the classification probability of each individual particle.

Note that during ACLOUD, one channel ($\theta = 34°$) was malfunctioning and is hence excluded from the analysis. During SOCRATES, the $\theta = 90°$ channel was observed be affected by the background noise in case of droplets and was thus excluded.

However, due to the design of the discrimination features (i.e. averaging over multiple channels) the implications on the discrimination are reduced and the same parameterization still works well for the SOCRATES data set.

## 3.5 Phase Discrimination using Machine Learning

Binary classification problems like the one presented in this work are typically well fit to be solved using machine learning (ML) algorithms (Kumari and Srivastava (2017)). For example, in recent works, Mahrt et al. (2019) and Touloupas et al. (2020) have presented different methods to employ ML to discriminate ice and liquid cloud particles using the PPD-HS and HOLIMO, respectively. Depending on the chosen classification problem, ML algorithms can be very easy and quick to set up: basically all one needs is a (pre-classified) training data-set. There exists software, such as e.g. *TensorFlow* (Google LLC, CA), that are specialised on ML, however, nowadays most common analysis software such as e.g. *Matlab* or *Mathematica* have built-in ML toolboxes that make working with ML quite easy, fast and comfortable. In general, the main idea is basically, that the ML algorithm is able to identify systematic differences and common features of the different "types" on its own (even such that could be hard to find for humans) and divide the data-set accordingly. This way, the ML can classify even new, unknown data-sets that "it has never seen" before. Given a large enough training data-set, ML algorithms can achieve high discrimination accuracies.

For comparison with the analytical approach used in this work, the classified data-set was analysed using two different, basic supervised ML methods, using a) fine decision tree and b) linear support-vector machine (SVM). This was done once for the raw data, i.e. just the scattering intensity of the 18 scattering channels (the $\theta = 34°$ and $\theta = 90°$ were removed) as well as using the four features $[f_1, f_2, f_3, f_4]$ presented in this work as well as using both raw intensity and derived features. Again, the data was trained using the ACLOUD data-set and tested against the SOCRATES data-set. All particles that had any missing values were discarded. The corresponding discrimination accuracies are shown in Tab. 1. It can be seen, that the different ML methods already show good results. Also, it shows once more that the presented features $[f_1, f_2, f_3, f_4]$ are indeed fit to represent the difference in the ASF. With more fine tuning, especially the discrimination accuracy of the SVM approach might reach the 99% of the analytical approach.

**Table 1.** Classification accuracies for different ML approaches and different input information.

| Used data-set | Fine Decision Tree | Linear SVM |
|---|---|---|
| Raw ASF data | 96.4% | 94.4% |
| Derived features | 97.9% | 98.4% |
| Both | 97.6% | 98.4% |

However, despite the discussed advantages, ML also has one main disadvantage: it is hard to understand what the algorithm is doing in detail. Basically, what you end up with, is a *black-box* that classifies input data with a given confidence, but you cannot tell why. Hence, it is very hard to analyze which features are relevant for the classification. Further, since the ML knows only statistics, not physics, it is possible that the ML algorithm links the classification to "un-physical parameters" that can introduce systematical biases. For example, it could be possible that the ML algorithm learns, that large particles (with a corresponding high total scattering intensity) are typically ice, whereas droplets are typically smaller and hence scatter less light. Thus, it would look at the "amplitude", rather than the "shape" of the ASF and classify all "large particles" as ice. Since

the number of large droplets in the used data-set is rather small, the overall discrimination accuracy would be quite high, however there would be the systematical bias that the few large droplets would tend to be misclassified.

Hence, and because it yields better discrimination accuracy, for this work, it was chosen to go with the "analytical approach" instead of ML. Also, the presented method has the advantage, as discussed previously, that it works without calibration for further campaigns, even when single scattering channels are malfunctioning (such as e.g. the $\theta = 34°$ channel during ACLOUD)
or the laser power is changed (since it takes only the shape, not the amplitude into account). Nevertheless, the presented analytical method works similar to the ML approach.

## 4 Particle Size Distribution

Since only a sub-sample of the PHIPS particle events produce a stereo micrograph (i.e. maximum imaging rate of 3 Hz in ACLOUD and SOCRATES), particle size distributions that are based on the analysis of the images can only be calculated with
275 a limited statistics. Furthermore, particle sizing might be biased for particles with sizes smaller than 30 µm, due to the limited optical resolution of the PHIPS imaging system (Schnaiter et al. (2018)). Hence, in the following section, particle sizing based on the single particle ASFs is introduced. The calibration based on the stereo micrographs is done following a similar approach as the phase discrimination in the previous section.

In order to calculate a particle number size distribution (PSD) per volume from the single particle sizing data, as shown in
Fig. 9, the volume sampling rate of the instrument has to be known. This sampling rate is simply the product between the speed of the aircraft and the sensitive area $A_{\text{sens}}$ of the trigger optics. The size of the sensitive area $A_{\text{sens}}$ is determined using optical engineering software. This is presented in section 4.2.

### 4.1 Particle Sizing

The individual detector channels of the PHIPS nephelometer measure scattered light intensity $I(\theta)$ of individual cloud particles
that can be converted to a differential scattering cross section, $\sigma_{\text{scatt}}^{\text{diff}}(\theta)$,

$$\sigma_{\text{scatt}}^{\text{diff}}(\theta) = I(\theta)/I_{\text{inc}} \cdot \pi \cdot d_{\text{laser}}^2/4, \tag{8}$$

with $I_{\text{inc}}$ and $d_{\text{laser}}$ the power and diameter of the incident laser beam, respectively. Note that $I(\theta)$ in Eq. (8) has to be corrected for possible background intensity due to stray light in the instrument as well as dark photon counts of the photomultiplier array. Integrating Eq. (8) over all nephelometer channels gives a partial scattering cross section, $\sigma_{\text{scatt}}^{\text{partial}}$, of the particle as defined for
the PHIPS measurement geometry

$$\sigma_{\text{scatt}}^{\text{partial}} = \pi \cdot d_{\text{laser}}^2/(4 \cdot I_{\text{inc}}) \cdot \int I(\theta)\, d\theta. \tag{9}$$

For spherical particles, $\sigma_{\text{scatt}}^{\text{partial}}$ is approximately proportional to their geometrical cross section $\pi \cdot D_{\text{p}}^2/4$, with $D_{\text{p}}$ the particle diameter. This is demonstrated in the supplementary material using Mie calculations (S2). Assuming that this is valid not only for spherical droplets but also for aspherical ice particles, the scattering cross section equivalent particle diameter $D_{\text{p}}^{\text{scatt}}$ can be

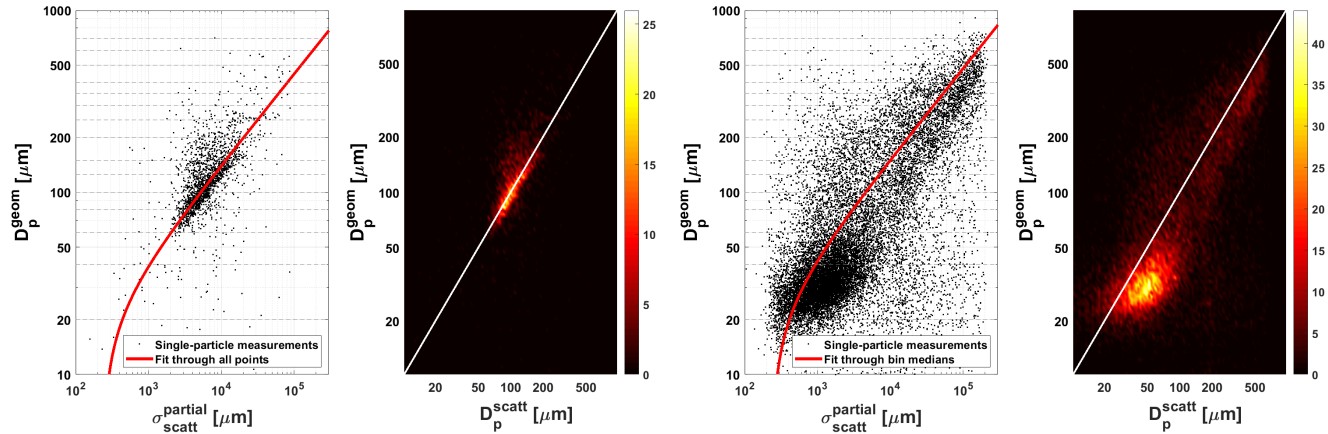

**Figure 8.** Calibration of the PHIPS integrated light scattering intensity measurement, expressed by the partial scattering cross section $\sigma_\mathrm{scatt}^\mathrm{partial}$, against the geometric diameter $D_\mathrm{p}^\mathrm{geom}$ deduced from the concurrent stereo micrographs. Stereo micrographs from the SOCRATES data set were manually classified for droplets (left) and ice particles (right).

deduced from the PHIPS intensity measurement $I(\theta)$

$$D_\mathrm{p}^\mathrm{scatt} = a \cdot \left( \int I(\theta)\, d\theta - c_\mathrm{BG} \right)^{\frac{1}{2}}. \tag{10}$$

In Eq. 10, $a$ is a calibration coefficient that describes the incident laser properties, the detection characteristics of the polar nephelometer (e.g. the photomultiplier gain settings) as well as the angular light scattering properties of the particle, and $c_\mathrm{BG}$ the integrated background intensity. As already discussed in the previous section, ice and droplets have vastly differing angular

scattering characteristics, i.e. scattering cross sections $\sigma_\mathrm{scatt}^\mathrm{diff}(\theta)$. Hence, different $a$ coefficients are needed and the calibration is done separately for ice and droplets. The coefficient $a$ is calibrated based on the geometric cross section equivalent diameter $D_\mathrm{p}^\mathrm{geom}$ derived from the stereo micrographs. A correction for the slight size overestimation of the CTA 2 for small particles due to the lower magnification is applied (see Schnaiter et al. (2018)). More details on PHIPS image analysis routines can be found in Schön et al. (2011).

Similar to the calibration of the phase discrimination algorithm, manually classified imaged particles were used as a calibration data set. The data is binned with respect to the particle's geometrical area equivalent diameter. The bin edges are the same as used for the final PSD data product. Those are 20, 40, 60, 80, 100, 125, 150, 200, 250, 300, 350, 400, 500, 600 and 700 µm. For ice, the coefficient $a$ is determined by fitting Eq. 10 through the median of each bin. For droplets, the function is fitted through all data points since the data points are distributed over fewer size bins. The background intensity $c_\mathrm{BG}$ is deter-

mined as the integrated intensity from forced triggers averaged over time periods when no particles were present. $c_\mathrm{BG}$ is the same for droplets and ice. The calibration is performed for each campaign separately, assuming that the instrument parameters remain unchanged over the duration of one campaign. The resulting calibration of the scattering equivalent diameter for the

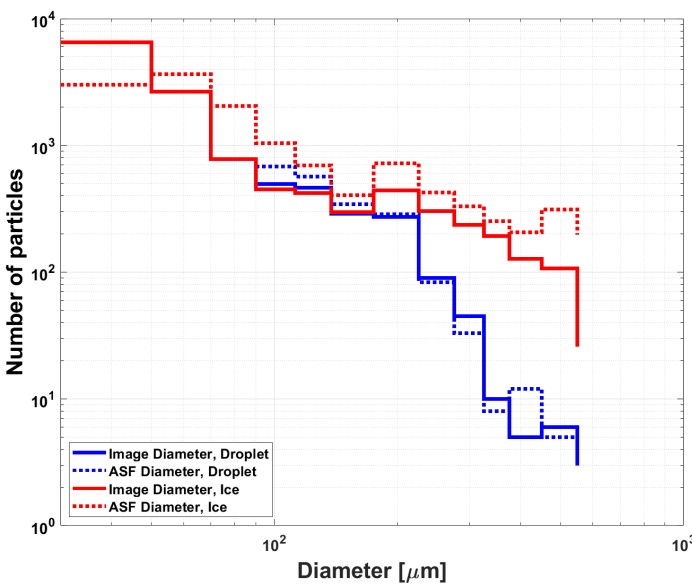

**Figure 9.** Comparison of PSD calculated from ASFs using calibration defined in Eq. 10 (dotted line) and PSDs based on $D_{\mathrm{p}}^{\mathrm{geom}}$ derived from stereo micrographs (average of CTA1 and CTA2, solid line) for droplets (blue) and ice particles (red). The data is from all flights recorded during SOCRATES. Only stereo micrographs that showed only one, completely imaged particle were taken into account. The same particles were used for both size distributions.

SOCRATES campaign is shown in Fig. 8a and Fig. 8b for droplets and ice, respectively. The corresponding fit parameters are $a_{\mathrm{ice}} = 1.4167$ and $a_{\mathrm{droplet}} = 1.4441$. The background measurement value is $c_{\mathrm{BG}} = 238.12$.

Using this calibration Fig. 9 shows the comparison of the particle size distributions averaged over all flights of SOCRATES for both ice (red) and droplets (blue). It can be seen that the size distribution based on the images (solid lines) agrees well with the size distribution based on the angular light scattering functions (dotted lines).

### 4.2 Sensitive Area

Due to the the facts that the scattering laser of PHIPS has Gaussian intensity profiles and the field of view of the trigger
optics shows gradual detection boundaries, $A_{\mathrm{sens}}$ is expected to be size dependent with a larger sensing area for larger particle sizes. Moreover, as (aspherical) ice particles usually have different differential scattering cross sections compared to (spherical) droplets, especially in side scattering directions where the trigger optics is located, $A_{\mathrm{sens}}$ is expected to be dependent also on the phase of the cloud particles. Therefore, we simulated the size dependence of $A_{\mathrm{sens}}$ for spherical and aspherical particles separately using the optical engineering software FRED (Photon Engineering, LLC, USA), which combines light propagation
by optical raytracing simulations with 3D computer aided design (CAD) visualization.

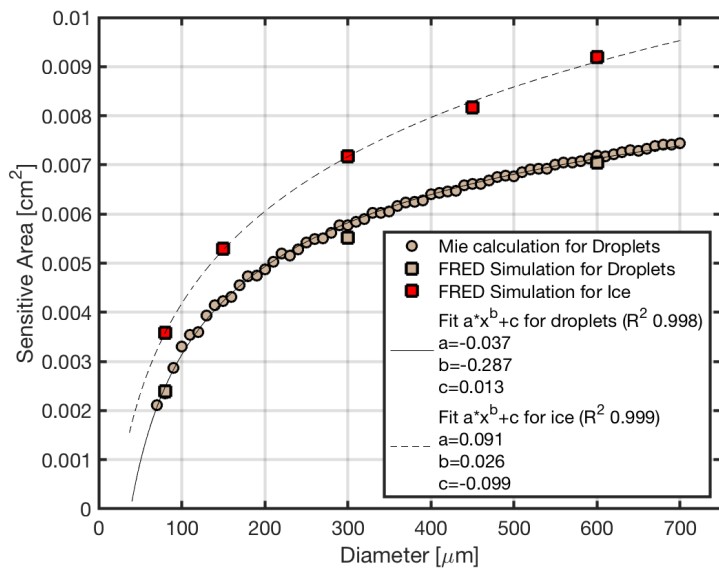

**Figure 10.** Sensitive area based on FRED simulations for ice (red) and droplets (grey)

For the FRED simulations, the actual PHIPS trigger optics and 3D laser intensity distribution were reconstructed in the 3D CAD environment of the software resulting in the actual intensity field the particle is exposed to when penetrating the sensitive area of the instrument. Particles were step-wise positioned at different x,y and z position across the trigger field of view and depth of field to get a map of the scattered light intensity that reaches the sensitive area of trigger detector. Similar to the actual measurement, a threshold value for the simulated detector intensity was used that would trigger the system and, therefore, defines $A_{\mathrm{sens}}$. This threshold was deduced by mapping the sensitive area of the instrument in the laboratory using a piezo-driven droplet dispenser which generates single 80 μm diameter water droplets (Schnaiter et al., 2018). Equating $A_{\mathrm{sens}}$ from the laboratory mapping with $A_{\mathrm{sens}}$ for the corresponding 80 μm FRED simulation then defined the threshold value that has to be used for all FRED simulations to calculate the size dependence $A_{\mathrm{sens}}$.

The FRED simulations were performed for spherical particles with the refractive index of supercooled liquid water ($n = 1.3362 + i1.82 \times 10^{-9}$) and the three sizes 80 μm, 300 μm and 600 μm. The resulting $A_{\mathrm{sens}}$ are shown in Fig. 10 in grey color. Additionally, to validate the method, $A_{\mathrm{sens}}$ was also estimated using Mie Theory to calculate the differential scattering cross section for the trigger direction and multiplying the results with the actual intensity field as defined by the FRED simulations. Although Mie calculations are faster to conduct, these calculations have the disadvantage that they assume a dimensionless particle, which induces uncertainties at the boundaries of the trigger field of view. Yet, the FRED simulations compare reasonably well with the results of the Mie calculations.

Ice particles were simulated roughened spheres whose surface light scattering was defined by the ABg Model (Pfisterer (2014)). A refractive index of $n = 1.3118 + i2.54 \times 10^{-9}$, Warren (1984)) was used for the ice simulations. The roughened ice

sphere approach was chosen here to avoid computationally expensive orientation averaging, which was necessary in case of

using a non-spherical particle habit. The FRED simulations for ice particles were conducted for the five particle sizes $80\,\mu m$,

$150\,\mu m$, $300\,\mu m$, $450\,\mu m$ and $600\,\mu m$. As can be seen in Fig. 10, the $A_{\text{sens}}$ values for ice are significantly larger than those

for water droplets of the same diameter. An exponential function was fitted to the FRED results to get $A_{\text{sens}}$ as a function of

particle diameter. These functional dependencies are then used to calculate the volume sampling rate that is required to convert

the single particle data to particle size distributions.

## 4.3   Correction for Shattering Artefacts

One major source of uncertainty for wing mounted probes is shattering of ice particles on the instrument's outer mechanical

structures or breakup of particles in the instrument inlet. An example of the shattering of a large particle and breaking up

of aggregates in the inlet flow field can be found in the supplementary material (S5). Shattering can lead to a significant

overcounting of ice particles (e.g. up to a factor of 5 using a fast forward scattering spectrometer probe (FSSP), Field et al.

(2003)) and a bias in the particle size distribution towards smaller sizes. Here, we characterized the frequency of shattering

events in the SOCRATES data set and present a method to detect shattering events within the PHIPS data sets. Even though the

geometry of PHIPS was designed to minimize disturbances and turbulences in the instrument (e.g. sharp edges at the front of

the inlet and an expanding diameter of the flow tube towards the detection volume (see Abdelmonem et al. (2016)), shattering

can still be an issue, especially in clouds where large cloud particles and aggregates with $D > 1$ mm are present.

Since the field of view of the camera telescope assembly (CTA) is much larger (typically $\simeq 1.5 \times 1$ mm) compared to the

sensitive trigger area (see previous section), the stereo micrographs can be used to detect shattering events. However, as only a

subset of detected particles is imaged, a shattering correction based on inspection of the stereo micrographs is not a practical

and reliable solution. Still, manual examination of the stereo micrographs can be helpful to determine whether or not a a cloud

segment was affected by shattering in individual cases.

### 4.3.1   Interarrival Time Analysis

The most common method to detect shattering that is based on the analysis of particle interarrival times Field et al. (2003). If

two (or more) particles are detected in very short succession, those particles are identified as shattering fragments and removed.

Fig. 11 shows a histogram of interarrival times ($\tau$) of ice particles (left) and droplets (right) measured during two flights of

SOCRATES. For ice, it is apparent, that the otherwise approximately log-normal distributed interarrival times show a second,

lower mode below $\tau \leq 0.5$ ms (equivalent to spatial separation of $\leq 7.5$ cm, assuming a relative air speed of $v = 150\,\text{ms}^{-1}$) that

is likely caused by shattering. For droplets, the second mode is not visible, since droplets tend to less fragment when entering

the instrument inlet.

Whereas the interarrival time analysis method is used in multiple optical array probes (2DS, 2DC, Field et al. (2003)), the

application is limited for single-particle instruments, like PHIPS, due to their small sensitive area. Near the detection volume,

the inlet has a diameter of $32\,\text{mm}$, whereas the sensitive area measures only about $0.7\,\text{mm}$ (depending on phase and size, as

discussed in 4.2), which means that the probability to detect two (or more) fragments of the same shattering event is very low.

Furthermore, the instrument has a dead time of $t = 12\,\mu s$ after each trigger event (Schnaiter et al., 2018). Shattering fragments that pass during this time, are not detected. As shown in Fig.11, only a small percentage of the particles whose images were

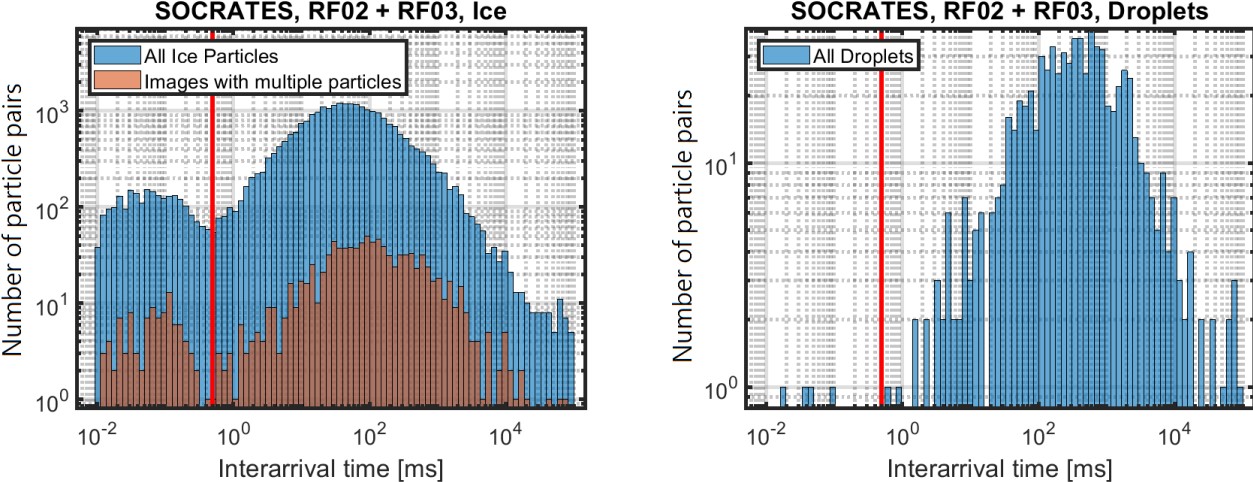

**Figure 11.** Histogram of interarrival times of ice particles (left) and droplets (right) measured during SOCRATES flights RF02 and RF03. Comparison of the interarrival times of all particles (blue) and only particles whose images were manually classified as shattering events (red). The red vertical line marks the $\tau \leq 0.5$ ms threshold.

manually classified as shattering (red), could be identified as shattering using the interarrival time analysis method. Hence it can be concluded, that interarrival time analysis alone is not fit as a reliable shattering flag, either. Nevertheless, all particles with a low interarrival time $\tau \leq 0.5$ ms are removed and excluded from the analysis. In the next section, a shattering flag is introduced, that flags segments which are affected by particle shattering, so they can be excluded from further analysis.

### 4.3.2 Shattering Flag based on the Presence of large Particles

It is known, that a particles shattering probability is strongly size dependent. Large particles and aggregates are much more prone to shattering compared to small particles. To overcome the limitation of the interarrival time method to eliminate shattered particles, we introduce a shattering flag based on the presence of large particles. Fig. 12a shows the total number concentration of particles in the size overlap region of PHIPS and 2DS ($200\,\mu m \leq D \leq 500\,\mu m$) for all SOCRATES flights. The data is averaged over 30s segments. Only segments with $N_{\text{2DS, overlap}} \geq 0.5\,\text{L}^{-1}$ are taken into account. The colour-code indicates the fraction of 2DS particles in the size range of $D_{max} \geq 200\,\mu m$ that are larger than $800\,\mu m$. The diagonal lines mark the median ratio between $N_{\text{PHIPS}}/N_{\text{2DS}}$ of each colour. Fig. 12b shows the correlation of the difference between PHIPS and 2DS in the overlap region and the ratio of large particles. It can be seen, that the two probes agree very well in segments with only a few large particles.

In segments that consist of more than 10% large particles, PHIPS and 2DS tend to disagree and PHIPS can overestimate particle concentrations up to a factor > 10. This can be explained by the shattering of large particles on the instrument inlet tip or wall or disaggregation of large aggregates due to shear forces in the inlet flow. Therefore, said marker for the presence of

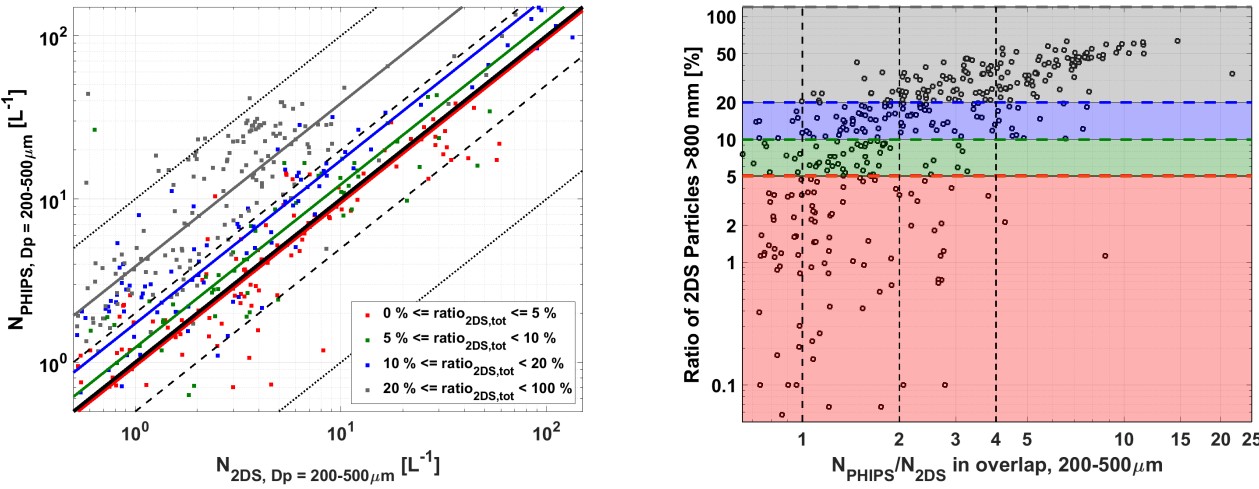

**Figure 12.** a) Comparison of the total number concentrations of 2DS and PHIPS. Each point is averaged over 30 s. The colour-code is based on the ratio of large 2DS particles with $D_{max} \geq 800\,\mu$m. The thick black line marks the 1:1 line, the dashed and dotted lines factor 2 and 10. b) Correlation of the ratio of number concentrations of PHIPS and 2DS and the presence of large 2DS particles. The horizontal line marks the 10% threshold. The colour-code is the same as in a).

large particles will be used as a shattering flag to mark cloud segments that are potentially affected by shattering. In segments where the 2DS did not detect any particles or was not measuring, for any reason, 2DC data is used instead. That means, cloud segments with more than 10% large particles are removed for future analysis. For the SOCRATES data set, 44% of all 1s segments are flagged as shattering. This means that about half of all 30s segments in mixed-phase clouds and approximately 400 75% of pure ice clouds are removed. Droplet dominated cloud segments are not affected by this shattering flag.

## 4.4 Discussion: Particle Size Distribution and Statistical Significance

The sampled cloud volume $V_s$ per unit time $t$ calculates as $V_s = A_{sens} \cdot v \cdot t$, where $v$ is the the relative air speed and $A_{sens}$ the probe's sensitive area. $A_{sens}$ is dependent on particle phase and diameter, as discussed in section 4.2. Assuming a relative air speed of $v = 150\,\mathrm{ms}^{-1}$, the resulting sample volume amounts about $V_s = 0.08\ (0.026, 0.12)\,\mathrm{L\,s}^{-1}$ for ice particles with 405 diameter D = 200 (50, 500) μm, respectively. This is somewhat larger compared to other single-particle cloud instruments (e.g. the CPI, $V_s = 0.009\,\mathrm{L\,s}^{-1}$, Lawson et al. (2001)), comparable to e.g. the SID3 ($V_s = 0.071\,\mathrm{L\,s}^{-1}$, Vochezer et al. (2016)), but is significantly smaller compared to the optical array probes like the 2DC ($V_s \simeq 0.1 - 10\,\mathrm{L\,s}^{-1}$, Wu and McFarquhar (2016)).

This has consequences for the averaging time needed in order to achieve statistically significant information on total particle concentrations.

We investigated the statistical uncertainty in example situations for the total number concentration for the size range from 20-200 µm. This size range was chosen since at sizes below 200 µm the phase information from PHIPS is of interest as phase detection based on traditional imaging methods can be challenging for small particle sizes. In order to reach statistical uncertainty $\propto n^{-0.5}$ of less than 10%, the number of particles per size bin need to be larger than n > 100. Table 2 shows the calculated averaging time in seconds that is needed until n = 100 particles are sampled per bin ($t_{n=100}$), the estimated number

of particles that would be sampled during 30 s of sampling ($n_{t=30s}$), as well as the corresponding statistical uncertainty $n^{-0.5}$ for a sampling period of 30 s ($\sqrt{n_{t=30s}^{-1}}$) for the chosen size range. All particles were assumed to be ice.

**Table 2.** Averaging time that is needed until n = 100 particles are sampled as well as the total number of particles sampled during an averaging time of 30 s, calculated for the size bin of 20-200 µm and exemplary particle concentrations.

| $D_{\text{lower edge}}$ | $D_{\text{upper edge}}$ | Concentration [L$^{-1}$] | $t_{n=100}$ [s] | $n_{t=30s}$ | $\sqrt{n_{t=30s}^{-1}}$ [%] |
|---|---|---|---|---|---|
| 20 | 200 | 1 | 1688.5 | 1.8 | 75.0 |
| 20 | 200 | 10 | 168.9 | 17.8 | 23.7 |
| 20 | 200 | 56.3 | 30 | 100 | 10 |
| 20 | 200 | 100 | 16.9 | 177.7 | 7.5 |
| 20 | 200 | 1000 | 1.7 | 1776.8 | 2.4 |

    It can be seen that the ice crystal concentrations need to be larger than $56.3\,\mathrm{L}^{-1}$ in order to achieve a statistical uncertainty below 10% within 30 s. For ice crystal concentrations of 1 (10) $\mathrm{L}^{-1}$ an averaging time of 28 (2.8) min would be needed, which at least in the case of low (< 10 L$^{-1}$) ice crystal concentrations would likely exceed the sampling duration. For optical

array probes assuming a sampling volume of $V_s = 1.5\,\mathrm{Ls}^{-1}$ the corresponding sampling times would be 66.7 s and 6.7 s for concentrations of 1 and 10 $\mathrm{L}^{-1}$. This shows, that in order to get statistically significant size distributions, it is important to properly consider adequate averaging time and/or bin size, especially in segments with low particle concentration.

## 5   Case Studies

In this section, the above presented methods are applied for three representative case studies from the SOCRATES campaign

in altitudes below 2000m, one purely liquid cloud and two mixed-phase clouds. The results are then compared to the measurements of other instruments from the same flights.

### 5.1   Case Study 1 - Pure Liquid Cloud

Fig. 13a shows meteorological and microphysical data collected during SOCRATES research flight RF04 on January 24th, 2018. Taking off in Hobart, Australia, the aircraft flew south-west sampling in different types of clouds ranging from deep

precipitating clouds to layer clouds in various different altitudes. The probing pattern was alternating between above cloud

sampling (for aerosol measurements) and in cloud sampling (to investigate the microphysical properties of the cloud's hydrometeors).

A low-level supercooled liquid cloud was probed in an altitude of approximately 2,100 m at a temperature of about -8.5°C at around 55°S, 141°E. The vertical wind velocity was at a constant value of $-0.5\,\text{ms}^{-1}$, indicating a weak downdraft. The relative humidity with respect to ice averaged about 105%. The liquid water content (LWC) measured with the CDP averaged around $0.1\,\text{g}\,\text{L}^{-1}$ and the total water content (TWC) measured with the 2DS was around $0.5\,\text{g}\,\text{L}^{-1}$. The lower panel shows the radar reflectivity measured by the HIAPER cloud radar (HCR, EOL (2018)), which shows a single non-precipitating cloud layer from 4:02 UTC onwards. The HCR beam was in nadir pointing mode for all three presented case studies.

The trigger threshold of PHIPS was set in a way that the instrument started to trigger on droplets with diameters larger than 50 μm. This remained unchanged over the whole campaign. The stereo micrographs from this flight segment (Fig.13c) show the presence of large drizzle droplets with diameters from 100 to 200 μm. No indication of the presence of ice crystals was seen in the PHIPS images.

Fig.13b shows PSDs measured with the CDP (UCAR/NCAR-EOL), 2DS (Wu and McFarquhar (2019)) and PHIPS. The PSD has a maximum at around 15 μm and the maximum particle sizes are found at 300 μm. All the PSDs agree well with each other. Information on the phase on the largest particles can be acquired from the PHIPS ASF measurements. The phase discrimination algorithm classified every particle in the presented segment as droplet, which is in agreement with the stereo micrographs. This shows, that this cloud, despite the low temperature and the particle sizes up to 300 μm, consists purely of supercooled liquid droplets.

## 5.2 Case Study 2 - Heterogeneous Mixed-Phase Cloud

Low-level mixed-phase clouds were investigated during SOCRATES research flight RF07 on January 31st, 2018. During this flight, the G-V sampled clouds south-east from Hobart, including an overpass over Macquarie island. The aircraft flew at cruising altitude towards the most southward point, where it descended down to lower altitude, probing multiple thin and persistent supercooled and mixed-phase clouds on its way back to Hobart.

Fig. 14a shows a cloud segment at around -58°N, 162°E, shortly after the turnaround at the most southward point. The cloud was probed in an altitude of 1,800 m at a temperature of about -10°C. The vertical wind velocity was slightly below zero and the relative humidity with respect to ice averaged about 107%. The maximum of the CDP LWC was $0.5\,\text{g}\,\text{L}^{-1}$ and the maximum of the 2DS TWC was $2\,\text{g}\,\text{L}^{-1}$.

Fig. 14b shows the PSDs between 04:16:40 and 04:21:00 UTC. The PSD has a maximum at 15 μm and the maximum particle sizes are found at 700 μm. All the probes agree well. Based on the PHIPS phase information, the whole segment can be divided in two sub-segments. Until 04:19:30 PHIPS detects only supercooled liquid droplets, after that only ice particles. This is backed up by PHIPS' representative stereo micrographs from the two sub-segments. In the first sub-segment, Fig. 14c shows supercooled drizzle droplets with diameters from 50-200 μm similar to the pure liquid case. During the second sub-segment Fig. 14d shows irregular and columnar ice crystals with sizes from 100-500 μm, some of which appear to be rimed or faceted. This coincides with the high reflectivity area measured by the HCR (lower panel in Fig. 14a) and the decrease in LWC

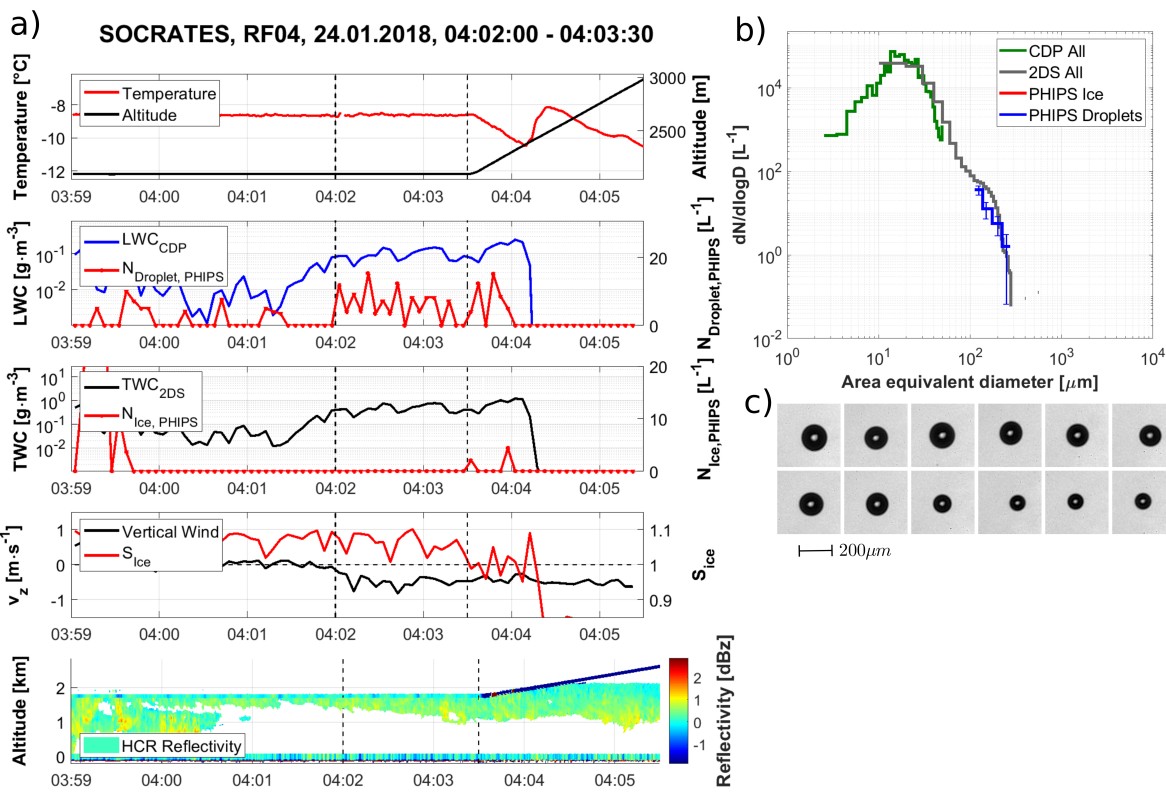

**Figure 13.** Example of PHIPS data acquired in a low-level supercooled liquid cloud over the Southern Ocean during the SOCRATES campaign (research flight RF04). a) overview of meteorological parameters, CDP, 2DS and PHIPS number concentrations (based on the ASF data) as well as HCR radar data. b) the comparison of the PSDs measured by CDP, 2DS and PHIPS including statistical uncertainty bars $\sqrt{n^{-1}}$ as discussed in 4.4. c) representative stereo micrographs of particles during that segment measured by PHIPS.

measured by the CDP. No ice particles were present on stereo micrographs taken during the first sub-segment and no droplets during the second, respectively.

### 5.3   Case Study 3 - Ice dominated Mixed-Phase Cloud

Fig. 15a shows a low-level mixed-phase cloud of SOCRATES research flight RF08 on February 4th, 2018. Due to a low pressure system south of Tasmania, cold air was expected advecting north from the Antarctic. During this flight, the aircraft flew straight southwards from Hobart. After turning back at the most southward point, the flight path back to Hobart was alternating between a "saw-tooth" pattern (going up and down through the clouds) and a "staircase" pattern (10 minutes above the cloud, then 10 minutes inside the cloud and 10 minutes below, as explained previously).

The presented case study shows one segment during the ascend of the final saw-tooth leg around -51°N, 147°E in a thin mixed-phase cloud in the Hallet-Mossop temperature regime (Hallet and Mossop (1974)). The cloud was approximately 700 m

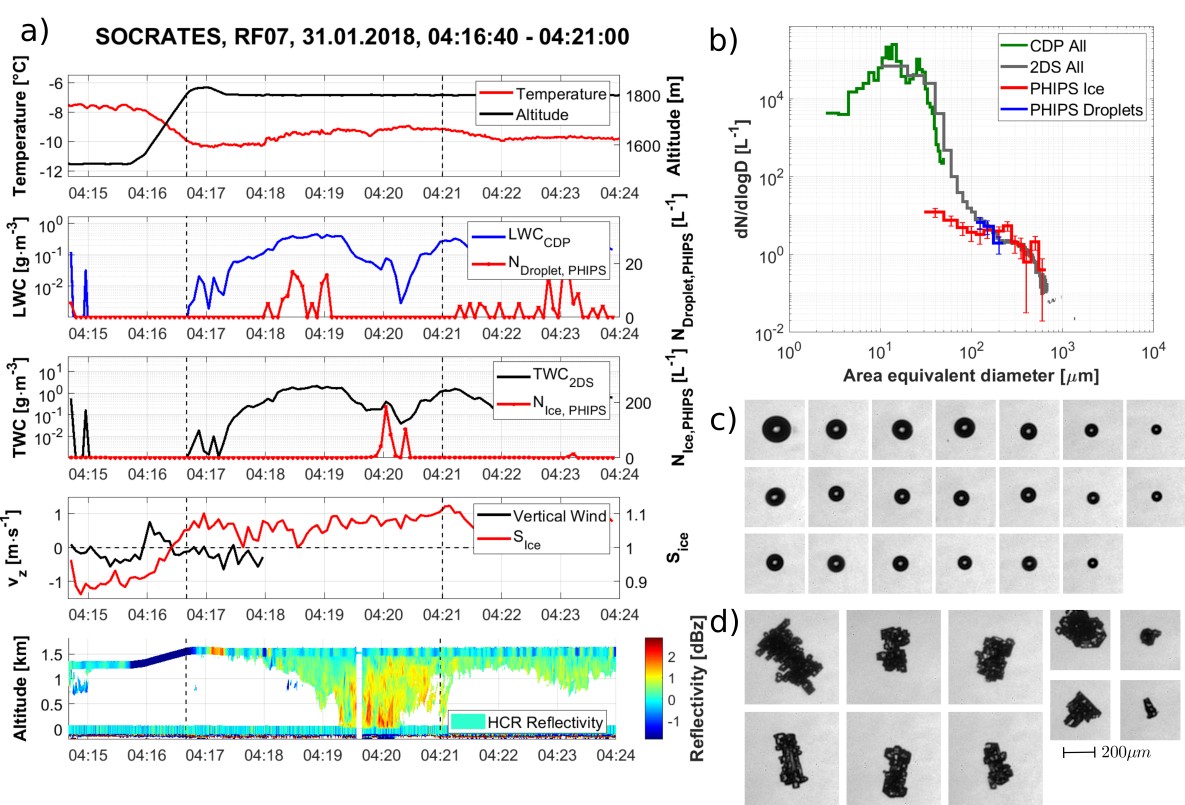

**Figure 14.** Same as in Fig.13 but for a low-level droplet-dominated mixed-phase cloud during a transit leg of SOCRATES research flight RF07. All supercooled droplets (c) were sampled between 04:16:40 - 04:19:30, whereas the ice particles (d) were sampled between 04:19:30-04:21:00.

thick and the temperature within the cloud ranged between -5°C at cloud base at 700 m and 0°C at the cloud top at 1400 m. The vertical wind velocity was fluctuating around zero and the relative humidity with respect to ice was between 80 and 100%. The maximum of the CDP LWC was $0.5\,\mathrm{g\,L^{-1}}$ and the 2DS TWC was $3\,\mathrm{g\,L^{-1}}$.

    Fig. 15b shows the PSDs between 05:13:10 and 05:15:35 UTC. The PSD has a maximum at 15 μm and the maximum particle sizes are found at up to 800 μm. Again, all three probes agree well. Contrary to the previous case, the stereo micrographs in

Fig. 15c+d are almost exclusively ice crystals. The sizes range from 20 μm to 500 μm. Observed ice crystal habits throughout the cloud were mostly needles with some hollow columns and small irregulars – all with different degrees of crystal complexity and riming. Also, a few supercooled droplets were present. The presence of supercooled droplets is also confirmed by the scattering measurements. This shows, that our method is also able to detect and correctly classify single large supercooled drizzle droplets in mixed-phase clouds which are otherwise dominated by ice in that size range.

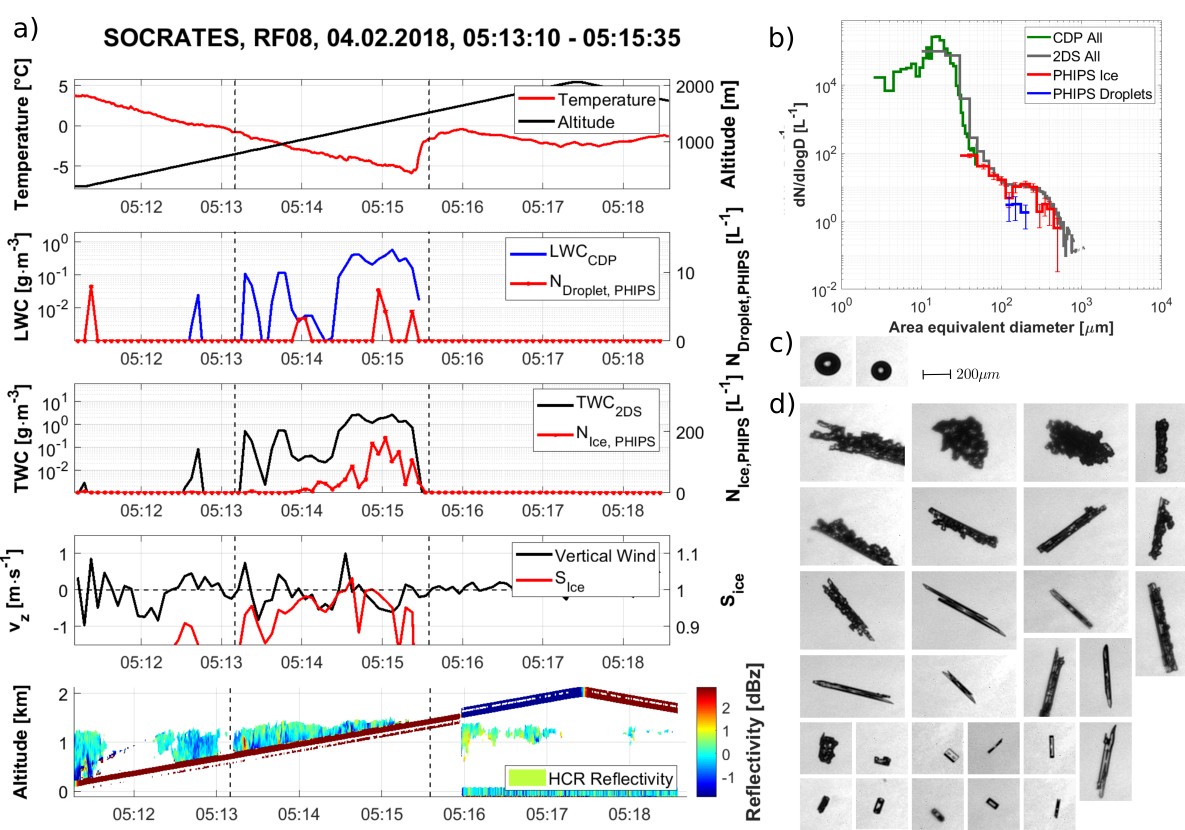

**Figure 15.** Same as in Fig.13 and Fig.14 but for a low-level mixed-phase cloud of SOCRATES research flight RF08.

## 6   Conclusions

A major challenge in the observations of mixed-phase clouds remains the phase discrimination of cloud droplets and ice crystals. Especially, in the size range of $D < 100\,\mu m$, reliable phase discrimination of cloud particles has been proven difficult. Here, we present a new method to derive the phase of single cloud particles using their angular light scattering information. ASFs of single cloud particles were measured with the airborne PHIPS probe. We identified four features in the particle light scattering function that were used for estimating the probability for the particle to be spherical or aspherical. The method was calibrated with a data set of 9,738 manually classified cloud particles and tested against a data set of 12,220 manually classified particles from two different aircraft campaigns. This yields a confidence rate above 98%. Further, we have shown that the phase discrimination algorithm is functioning independently of the experimental data set used for the calibration, so no further calibration is needed for upcoming future campaigns.

Additionally, we presented a method to derive PSDs based on single particle scattering data for particles in a size range from $100\,\mu m \leq D \leq 700\,\mu m$ and $20\,\mu m \leq D \leq 700\,\mu m$ for droplets and ice particles, respectively. The newly developed data analysis

algorithms were applied to three case studies that did not show the presence of large (>1 mm) ice crystals. Comparison of the PSDs from other instruments showed a good agreement. The presented case studies show, that PHIPS can provide unique and detailed insight about the phase composition of clouds, where phase discrimination based solely on particle size or aspect ratio could potentially be difficult, such as e.g. in mixed-phase cloud conditions where large droplets and small ice crystals coexist.

With these methods available, PHIPS can provide additional information on the microphysical properties of mixed-phase clouds in situations, where the data is not affected by shattering. We have also shown that phase discrimination based on single-particle angular light scattering behaviour is a robust method, which could be implemented in future cloud research instrumentation.

*Code availability.* The code used for the phase discrimination is available at https://doi.org/10.5281/zenodo.4321316.

*Data availability.* The PHIPS single particle scattering data can be found online in the PANGEA database (https://doi.org/10.1594/PANGAEA.902611) for ACLOUD and the EOL database (https://doi.org/10.5065/D6639NKQ) for SOCRATES. The single particle microscopic stereo images are available upon request from the authors.

*Author contributions.* FW, EJ and MS developed the ice discrimination algorithm, particle sizing calibration and shattering correction. EJ and MS collected the PHIPS data from the ACLOUD campaign. EJ, MS and FW collected the PHIPS data from the SOCRATES campaign. EJ and FW analysed the PHIPS data. MS conducted the optical engineering for the updated simulated sensitive area of PHIPS. FW wrote the manuscript with help from EJ and MS. All were involved in the discussion and commented on the paper.

*Competing interests.* The authors declare that they have no conflict of interest.

*Acknowledgements.* We express our gratitude all participants of the field studies for their efforts, in particular the technical crew of the AWI Polar 6 and NSF G-V. We would like to acknowledge operational, technical and scientific support provided by NCAR's Earth Observing Laboratory, sponsored by the National Science Foundation. We thank Wei Wu for valuable discussions. We would also like to thank the technical and scientific staff of IMK-AAF for their continuous support. This work has received funding from the Helmholtz Research Program Atmosphere and Climate and by the German Research Foundation (DFG grant JA 2818/1-1).

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

## Appendix A: Phase Discrimination Algorithm: Fit Parameters

The to fit the normalized occurrence of the feature parameters in Fig. 6 (upper panels), a Gaussian fit function of the form

$$n(x) = a \cdot \exp\left(-\left(\frac{x-\mu}{\sigma}\right)^2\right) \tag{A1}$$

is used. The corresponding fit parameters (with 95% confidence intervals) for the four feature parameters for the ACLOUD data set are shown in Tab. A1.

**Table A1.** Fit parameters of the Gaussian fits for the distribution of the feature parameters $n_i$.

| Feature | $a_{\text{Droplet}}$ | $\mu_{\text{Droplet}}$ | $\sigma_{\text{Droplet}}$ | $a_{\text{Ice}}$ | $\mu_{\text{Ice}}$ | $\sigma_{\text{Ice}}$ |
|---------|---------|---------|---------|---------|---------|---------|
| Mie | 150.3 ($\pm$7.2) | 3.842 ($\pm$0.011) | 0.3184 ($\pm$0.0167) | 868.5 ($\pm$18.7) | 5.195 ($\pm$0.004) | 0.2451 ($\pm$0.0059) |
| Down | 198 ($\pm$5.6) | -9.444 ($\pm$0.029) | 1.243 ($\pm$0.041) | 331.6 ($\pm$13) | -2.413 ($\pm$0.101) | 3.137 ($\pm$0.131) |
| Up | 62.48 ($\pm$4.68) | 9.348 ($\pm$0.23) | 3.592 ($\pm$0.287) | 261 ($\pm$14.5) | 1.789 ($\pm$0.149) | 3.299 ($\pm$0.211) |
| Minimum | 127.9 ($\pm$3.8) | -1.553 ($\pm$0.009) | 0.3859 ($\pm$0.0134) | 406.2 ($\pm$14.7) | -0.1919 ($\pm$0.0143) | 0.492 ($\pm$0.0105) |

Since the Gaussian distributions are of similar width $\sigma$, the corresponding discrimination probabilities (Fig. 6, lower panels),
defined as

$$P(f) = \frac{n_{\text{ice}}(f)}{n_{\text{ice}}(f) + n_{\text{droplet}}(f)}, \tag{A2}$$

can be approximated by a sigmoid function of the form

$$P(x) = \frac{a-d}{1 + \exp\left(-b \cdot (x-c)\right)} + d. \tag{A3}$$

The corresponding fit parameters are shown in Tab. A2.

**Table A2.** Fit parameters of the sigmoid fit for the discrimination probabilities $P_i$.

| Feature | $a$ | $b$ | $c$ | $d$ |
|---------|-----|-----|-----|-----|
| Mie | 98.57 ($\pm$0.83) | 10.89 ($\pm$0.57) | 4.641 ($\pm$0.006) | 0.1804 ($\pm$0.6129) |
| Down | 99.36 ($\pm$1.02) | 2.52 ($\pm$0.317) | -7.312 ($\pm$0.057) | 2.052 ($\pm$1.369) |
| Up | 98.04 ($\pm$1.26) | -1.069 ($\pm$0.096) | 5.732 ($\pm$0.097) | 3.14 ($\pm$1.036) |
| Minimum | 99.27 ($\pm$1.73) | 10.78 ($\pm$2.319) | -0.9897 ($\pm$0.023) | 2.194 ($\pm$2.234) |

## Appendix B: Phase Discrimination Algorithm: Cross-correlation of the Feature Parameters

In section 3.3 we have argued, that one feature alone is not sufficient to reliably classify all cloud particles, due to the particles that lie in the overlap between the two peaks in Fig. 6. Now the question is, how dependent the four features are and whether

or not a particle, that cannot be confidently (or is even falsely) classified by e.g. $f_3$, i.e. that lies in the overlap of the feature space, can be confidently classified by the other feature parameters or if it lies in the overlap for the other features as well.

Figure B1a shows the correlation of the classification confidence based on only one feature parameters $f_3$ and of the combined result for all 4 features for all manual classified ice particles of the ACLOUD campaign. It can be seen, that lots of particles that cannot be classified with high confidence by the first feature ($P(f_3) < 66\%$) are classified with high confidence by the other features ($P_{combined} > 66\%$). The corresponding statistics are displayed in a confusion matrix in Fig. B1b. It can be seen, that most of the particles (87.5%) are correctly and confidently classified based on $f_3$ alone (column 4). But out of the 992

particles that are not classified confidently and correctly based on $f_3$ (i.e. sum of column 2 and 3) most (805) are confidently classified based on the combination of all four features. This shows, that the usage of multiple features significantly improves the discrimination accuracy. Hence, by combining all four different features, a high combined classification confidence can be achieved as shown in Fig. 6a in the SI.

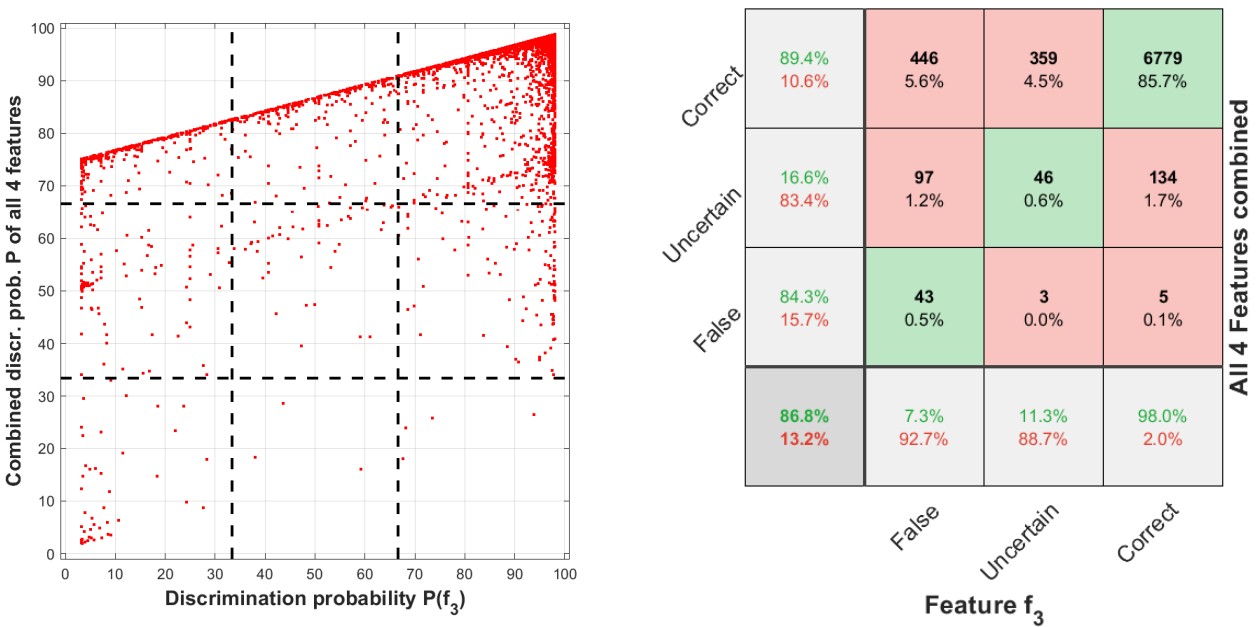

**Figure B1.** a) Correlation of the classification probability of feature parameters $f_3$ alone and the combination of all four features. The dashed lines mark the confidence limits. $P(f) > 66\%$ corresponds to particles, that are classified correctly with high confidence, $33 < P(f) \leq 66\%$ means the classification is uncertain and particles with $P(f) \leq 33\%$ are classified falsely as droplets with high confidence. b) shows the corresponding statistics of the plot in a confusion matrix. The squares correspond to the dashed lines in a).