# Peer review of "PHIPS-HALO: the airborne particle habit imaging and polar scattering probe - Part 3: Single Particle Phase Discrimination and Particle Size Distribution based on Angular Scattering Function"

_Atmospheric Measurement Techniques, 2020_

## Referee Comment (RC1) · Anonymous Referee #2 · 9 Oct 2020

Classification and phase discrimination of cloud particles, especially of mixed phase clouds, are of importance in a number of applications: modelling of the earth radiative balance and the clouds life cycle, interpretation of remote sensing data and so on. Phase discrimination and classification along with size estimation are usually performed using particles images. Several approaches and algorithms were reported in the literature and showed a good performance when applied to particles images. At the same time, based-on-images discrimination between droplets and quasi-spherical ice

particles is an extremely challenging task. It is well known that there exist significant differences between phase functions of water droplets and atmospheric ice particles. That fact was proved in a large number of modeling works. It was confirmed in experimental works where angular scattering intensities were measured in situ. The advantage of the PHIPS-HALO probe consist in the fact that a particle stereo-image and the corresponding angular distribution of the scattered light are recorded simultaneously. The synergy of those data provides significant improvement of the discrimination quality.

The work under reviewing addresses relevant scientific questions it is within the scope of AMT. I recommend that the paper be published in AMT after minor revisions.

Specific comments:

Figs. 1, 2 and 5; page 4 line 15; page 5 lines 11 – 14. Single spherical particles, the authors are dealing with, have the size parameter of 590 or higher. Phase functions of large spheres can be found in numerous textbooks and they differ much from "theoretical scattering functions" shown in Figs. 1-2. Mie calculations are mentioned several times in the text of the preprint before it is underscored (line 16 of the page 6) that "the calculated theoretical Mie scattering is integrated over the field of view of the polar nephelometer channels". Such important point should be underscored at the first mention of Mie calculations. And, I believe that the data from the light scattering databases by Baum et al. (2011) and Yang et al. (2013) were integrated over the field of view as well.

Fig. 1. It is written: "SOCRATES, RF02, #613, Spherical Ice". The particle shown in in Fig. 1b is not spherical; I would say it is quasi-spherical. Moreover, to my knowledge, there are no spherically symmetric particles that are able to provide such kind of the angular scattering function (ASF) as the red curve in Fig. 1c. The surface roughness and/or small internal inclusions cannot lead to an ASF that is increasing within the range [42 – 74] degrees. In my opinion, that ASF is the outcome of the deviation from

the spherical symmetry. If the authors can provide another explication, it would be useful to see it in the paper text.

page 5 lines 11 – 23. That part of the text should be revised. It is very difficult to understand how "the first discrimination feature f1" is computed even for an experienced reader. If I have understood correctly, the first step is the normalization of EVERY measured ASF by the ASF that corresponds to the spherical particle with the diameter of 100 $\mu$m. Next. What does it mean "the median over all channels"? How it is computed? Next. If the meaning of the "feature f1" is "the deviation of the observed ASF from the calculated Mie scattering", why it has such high values for spheres as in Figs. 4a and 5a?

Figs. 4a and 5a Why the Gaussian fit of the feature f1 for droplets in Fig. 4a has the mean value (about 3.8) that differs much from the value (about 2.5) in Fig. 5a?

Section 3.2 The PHIPS-HALO provides ASFs for a particle that has random but fixed orientation in the space. To my knowledge, the databases from Baum et al. (2011) and Yang et al. (2013) provide scattering properties averaged over random orientations of particles. If so, Fig. 5 only shows that the proposed method is not in contradiction with properties of ensembles of ice particles.

Section 3.3 I would say that the calibration-and-verification approach, the authors used, is somewhat similar to methods of the neural networks. Of course, the choice of parameters in the work under reviewing is well grounded and corresponds to general features of scattering by spherical and non-spherical particles. At the same time, it would be interesting to see in future works comparison with performance of neural networks algorithms.

page 18 line 11. The HIAPER cloud radar is capable of collecting observations in a staring mode between zenith and nadir or in a scanning mode. Thus, it is worth mentioning in the text that the HCR beam was in nadir pointing mode for all Case Studies of Section 5.

**[AMTD](page-top-right)**

Supplement (Fig. 9) In my opinion, the measured ASF differs much from the Mie calculation, especially in the range of [18 – 50] degrees. Nevertheless, the algorithm misclassified it. Thus, some improvements of the authors' approach can be done in the future.

———————————————————

---

## Referee Comment (RC2) · Greg McFarquhar (Referee) · 25 Oct 2020

Review of "PHIPS-HALO: the airborne particle habit imaging and polar scattering probe- Part 3: Single particle phase discrimination and particle size distribution based on angular scattering function" by Waitz et al.

Recommendation: Accept with minor revision

[Figure]

This study uses data collected by the PHIPS-Halo during two field campaigns (ACLOUD and SOCRATES) to develop a method to determine the phase of individual cloud hydrometeors from the light scattering measurements made by the probe. The algorithm takes advantage of differences in features between the angular scattering of spherical and aspherical particles to show that it can be determined with a 98% accuracy whether the particles are liquid or ice (i.e., spherical or aspherical). They also present a method for deriving the particle size distributions from the measured data. Information about particle phases is desperately needed because mixed-phase clouds are still not well understood. Further, the full potential of the PHIPS-HALO probe has yet to be realized because both particle size distributions and particle phase distributions have not been routinely been made available in a short time period after the conduct of field campaigns. As the material in this paper works to overcome both of these shortcomings, it should be published as soon as possible and represents a good contribution to the refereed literature. The paper is well written and technically sound so does not require a lot of revisions in that respect. Nevertheless, I think there are a couple of aspects that should be better explained in the paper so that the limitations, as well as the strengths, of the PHIPS-HALO for providing size and habit information are well outlined.

My major critique of the paper is that I think more information about the statistical representativeness of the data that are available from the PHIPS HALO probe should be included. The sample volume and/or sample area of the PHIPS-HALO probe should be explicitly stated. How does that compare against the sample volume/area from the commonly used. How does that affect the averaging time over which representative particle size distributions and particle phase distributions are available? For example, McFarquhar et al. (2007) calculate the required time that particle size distributions would need to be averaged over in order to obtain statistically significant particle size distributions that they defined to mean 100 particles in each size bin (so that there was a 10% uncertainty assuming the statistical uncertainty was proportional to the square root of the number of counts in each size bin). Figure 9 defines the sensitive area of

the PHIPS-HALO as ranging from 0.002 cm^2 to about 0.01 cm^2 depending on the size of the particle. Assuming a roughly 150 m/2 air speed, this would give sample volumes ranging from 30 to 150 cm^3 per second, or about 0.15 liter/second. This would seem to be quite a bit less than that of the 2DS/2DC class of probes. This does not negate the benefits of the PHIPS-HALO probe, but rather would seem to suggest that the more detailed phase/shape information available from the PHIPS-HALO probe has synergy with the more frequent data available from the optical array type probes that are better suited for deriving the fine resolution structure of clouds. This point, including explicitly comparing the sampled areas/volumes and numbers of particles between probes should be explicitly shown in the paper, and also mentioned in the abstract.

Detailed Comments

Page 1, line 6: evaluated would be a better word than validated. Page 2, line 13: "a large sampling statistics is required" reads awkwardly and should be rephrased. Page 3, line 15: the plural of aircraft is aircraft. Page 4, line 7: Suggest adding Um et al. 2011 ACP to the list of references as they considered scattering functions of several models of quasi-spherical ice crystals Page 4, Figure 2: Can you state what are the maximum dimensions of the two ice crystals that are considered in the figure? Page 7, line 23: It should be noted and discussed why there are a lot of differences in the nature of the distributions between the observed and modeled particles in Figures 4 and 5. Does this suggest that there are some limitations in how well the theoretical models are representing the actual observed particles? Page 9, line 3. This may be a stylistic thing, but when I see 41.000 I think there are 41 particles. I think the authors mean 41,000. Unless this convention is demanded by ACP, I would use a comma rather than a period. Page 9, line 3: Related to my major point above, I think it would be very interesting to compare the number of particles that were measured by the 2DC/2DS for the same periods during these field projects. That would help clarify information about the statistical representativeness of the data. Page 11, Figure 6.

Was any effort made to go back and look at the particles that were misidentified to determine why they were misidentified? I agree that 98% is outstanding (and better than classifications that are based on other probe data), but it would still be interesting to know why the discrepancy for just these few particles. Was there any chance that the manual identification of these particles was incorrect? Page 13, Figure 8. Most probes that measure small particles have smaller sensitive areas for smaller particles than larger particles (e.g., see Figure 9 in this paper). But, as I understand it the vertical axis here is number of particles rather than some measure of concentration per bin or number distribution function. It would be informative to include another plot that shows the calculated number distribution function since that is a physically meaningful quantity, especially since the caption reads that this is a particle size distribution (PSD). Page 18, Figure 11. The caption should specifically state the averaging period for which each of the plotted points corresponds to. Page 22, Line 10. To make this study more accessible, it would be nice to have the codes used available on github or some other code repository.

---

## Author Comment (AC1) · 18 Jan 2021

We thank the anonymous reviewer for his/her helpful comments. These comments helped to substantially improve the manuscript. Below we give detailed answers to the reviewer's comments that are highlighted in *cursive*.

Classification and phase discrimination of cloud particles, especially of mixed phase clouds, are of importance in a number of applications: modelling of the earth radiative balance and the clouds life cycle, interpretation of remote sensing data and so on. Phase discrimination and classification along with size estimation are usually performed using particles images. Several approaches and algorithms were reported in the literature and showed a good performance when applied to particles images. At the same time, based-on-images discrimination between droplets and quasi-spherical particles is an extremely challenging task. It is well known that there exist significant differences between phase functions of water droplets and atmospheric ice particles. That fact was proved in a large number of modeling works. It was confirmed in experimental works where angular scattering intensities were measured in situ. The advantage of the PHIPS-HALO probe consist in the fact that a particle stereo-image and the corresponding angular distribution of the scattered light are recorded simultaneously. The synergy of those data provides significant improvement of the discrimination quality. The work under reviewing addresses relevant scientific questions it is within the scope of AMT. I recommend that the paper be published in AMT after minor revisions.

*We thank the reviewer for this encouraging general comment. Below we have addressed the proposed minor revisions.*

Specific comments: Figs. 1, 2 and 5; page 4 line 15; page 5 lines 11 – 14. Single spherical particles, the authors are dealing with, have the size parameter of 590 or higher. Phase functions of large spheres can be found in numerous textbooks and they differ much from "theoretical scattering functions" shown in Figs. 1-2. Mie calculations are mentioned several times in the text of the preprint before it is underscored (line 16 of the page 6) that "the calculated theoretical Mie scattering is integrated over the field of view of the polar nephelometer channels". Such important point should be underscored at the first mention of Mie calculations. And, I believe that the data from the light scattering databases by Baum et al. (2011) and Yang et al. (2013) were integrated over the field of view as well.

*An explanation that the theoretical scattering data is integrated over the field of view of each nephelometer channel was added in Fig. 1 when Mie calcula-*

*tions are first mentioned. Also, it is now noted again in 3.2, that the scattering data from Baum et al. (2011) and Yang et al. (2013) is also integrated over the nephelometer channel field of view. A subsection explaining the integration of the ASF over the field of view of the nephelometer channels was added in the supplementary information (S6).*

Fig. 1. It is written: "SOCRATES, RF02, #613, Spherical Ice". The particle shown in in Fig. 1b is not spherical; I would say it is quasi-spherical. Moreover, to my knowledge, there are no spherically symmetric particles that are able to provide such kind of the angular scattering function (ASF) as the red curve in Fig. 1c. The surface roughness and/or small internal inclusions cannot lead to an ASF that is increasing within the range [42 – 74] degrees. In my opinion, that ASF is the outcome of the deviation from the spherical symmetry. If the authors can provide another explication, it would be useful to see it in the paper text.

> *The reviewer is correct that a spherical ice would not result into the measured ASF, and thus, the caption was corrected to "quasi-spherical ice".*

page 5 lines 11 – 23. That part of the text should be revised. It is very difficult to understand how "the first discrimination feature $f_1$" is computed even for an experienced reader. If I have understood correctly, the first step is the normalization of EVERY measured ASF by the ASF that corresponds to the spherical particle with the diameter of 100 µm.

> *We agree, this part of the text was difficult to understand. Hence, we have added a step-by-step explanation of the determination of the $f_1$ parameter based on two exemplary droplets including graphical explanation (Figure 4) of the in-between-products $q$, $\bar{q}$ and $q$'. This should make it easier to understand.*

Next. What does it mean "the median over all channels"? How it is computed?

> *The "median over all channels", $\bar{q}$, is calculated as the median of all values*

$q(\theta)$ of each channel $\theta$. $q(\theta)$ is the ratio of the measured scattering intensity
$I_{exp}(\theta)$ to the theoretical Mie calculation $I_{Mie}(\theta)$ for every channel $\theta$.

Next. If the meaning of the "feature f1" is "the deviation of the observed ASF from the calculated Mie scattering", why it has such high values for spheres as in Figs. 4a and 5a?

> The "Mie-comparison-feature" $f_1$ is based on the relative difference between the measured and calculated ASF of a reference Mie-sphere with diameter D=100 μm. By definition, the $f_1$ value for a simulated droplet with D=100 μm, of course, is zero since basically the input equals the reference, i.e. $I_{Exp}$ = $I_{Mie}$. However, this method is sensitive to small deviations from the theoretical curve (i.e. the Mie calculation for D=100 μm). For example, if you take the calculated $I_{Mie}$(D=100 μm) and alter the intensity of every other channel by +/- 2%, the resulting $f_1$ value = 1.1. Further, the difference between the shape of Mie calculations for different diameters is very small, but non-vanishing. For example, for a Mie-sphere with D=200 μm has $f_1$= 2.2. For actual, in-situ measured droplets, this deviation can be even larger due to deformation and impurities. E.g., the exemplary measured droplet shown in Fig. 2 has $f_1$=3.7. This example is now also discussed in section 3.1.1.

Figs. 4a and 5a Why the Gaussian fit of the feature f1 for droplets in Fig. 4a has the mean value (about 3.8) that differs much from the value (about 2.5) in Fig. 5a?

> The $f_1$ parameter value, i.e. the difference between Mie calculation and measured ASF, is supposed to be quite small for simulated particles (as the reviewer rightfully pointed out in her/his previous comment). For measured droplets, the difference can be slightly higher due to fluctuations or slight deformation, as already mentioned in the previous comment. Hence this relatively large discrepancy and shift towards higher values is to be expected, compared to the good agreement of the other features. However, we agree

*with the reviewer, that a detailed comparison and discussion of the feature-parameter-distribution-plots is missing. This was added in section 3.3: "The plots show that the distribution of the four aforementioned feature parameters are clearly distinct for droplets and ice and thus represent features that can be used to discriminate droplets from ice. Further, it can be seen that these normalized occurrences ($f_i$) are normally distributed. The distributions of the four feature parameters based on the measurements (Fig. 6) show a similar trend to the simulations (Fig. 5). The width of the distributions of feature parameters for measurements is much broader compared to the simulations. This can be explained by the single-orientation of the measured crystals compared to the orientation-averaging that was used in the simulations. Orientation-averaging tends to smooth out features in the ASFs and thus cause more narrow feature parameters. It should be also noted that the theoretical computations are for idealised crystals. Nevertheless, the mean values of the distributions agree very well. The only exception to this is the mean value of the distribution of droplets for f1, which is shifted slightly to larger values compared to the simulations. This is to be expected because the "Mie-comparison-feature" $f_1$ is based on the relative difference between the measured and calculated ASF. This difference is much smaller for simulated particles as discussed in 3.1.1."*

Section 3.2 The PHIPS-HALO provides ASFs for a particle that has random but fixed orientation in the space. To my knowledge, the databases from Baum et al. (2011) and Yang et al. (2013) provide scattering properties averaged over random orientations of particles. If so, Fig. 5 only shows that the proposed method is not in contradiction with properties of ensembles of ice particles.

*This is true. This is noted in the added discussion comparing the simulated and measured feature-parameter-distribution-plots mentioned in the answer to the previous comment.*

Section 3.3 I would say that the calibration-and-verification approach, the authors used,

is somewhat similar to methods of the neural networks. Of course, the choice of parameters in the work under reviewing is well grounded and corresponds to general features of scattering by spherical and non-spherical particles. At the same time, it would be interesting to see in future works comparison with performance of neural networks algorithms.

*Yes, the calibration-and-verification approach is quite similar to the approach of neural networks or machine learning algorithms. Classification using machine learning, both based on either the raw ASF data as well as the derived features [f1,f2,f3,f4], was tried. The classification accuracy was almost as good (96.4-98.4%, depending on the used algorithm) as the "analytical approach" presented in this work. However, machine learning also has one main disadvantage: it is hard to understand what the algorithm is doing in detail. Basically, what you end up with, is a "black-box" that classifies input data with a given confidence, but you cannot tell why. Hence, it is very hard to analyse which features are relevant for the classification. Further, since the machine learning knows only statistics, not physics, it is possible that the machine learning algorithm links the classification to "un-physical parameters" that can introduce systematical biases. For example, it could be possible that the machine learning algorithm learns, that large particles (with a corresponding high total scattering intensity) are typically ice, whereas droplets are typically smaller and hence scatter less light. Thus, it would look at the "amplitude", rather than the "shape" of the ASF and classify all "large particles" as ice. Since the number of large droplets in the used data-set is rather small, the overall discrimination accuracy would be quite high, however there would be the systematical bias that the few large droplets would tend to be misclassified. Hence, and because it yields better discrimination accuracy, for this work, it was chosen to go with the "analytical approach" instead of machine learning. The results of the machine learning as well as a detailed discussion are now included in section 3.5.*
page 18 line 11. The HIAPER cloud radar is capable of collecting observations in a staring mode between zenith and nadir or in a scanning mode. Thus, it is worth mentioning in the text that the HCR beam was in nadir pointing mode for all Case Studies of Section 5.

*A note was added in section 5 that the HIAPER cloud radar was in nadir pointing mode for all case studies.*

Supplement (Fig. 9) In my opinion, the measured ASF differs much from the Mie calculation, especially in the range of [18 − 50] degrees. Nevertheless, the algorithm misclassified it. Thus, some improvements of the authors' approach can be done in the future.

*The first channels are often times saturated and hence are not taken into account. The rest of the ASF for this particular particle looks quite similar to a droplet's. Approaches that only exclude the first channels if they are saturated and include them otherwise, were tried and could be able to correctly classify particles as this one, but resulted in an overall decrease of the discrimination accuracy (i.e. particles that are classified correctly now would then be misclassified). The algorithm was calibrated to optimize the overall discrimination accuracy, i.e. that the highest fraction of particles is classified correctly. Nevertheless, there might be quite possibly be more room to improve the presented algorithm in the future with more data from future campaigns. This discussion was also added in the SI.*

---

## Author Comment (AC2) · 18 Jan 2021

> We thank the Greg McFarquhar for his helpful comments. These comments helped to substantially improve the manuscript. Below we give detailed answers to the individual reviewer comments in *cursive*.

This study uses data collected by the PHIPS-Halo during two field campaigns (ACLOUD and SOCRATES) to develop a method to determine the phase of individual cloud hydrometeors from the light scattering measurements made by the probe. The algorithm takes advantage of differences in features between the angular scattering of spherical and aspherical particles to show that it can be determined with a 98% accuracy whether the particles are liquid or ice (i.e., spherical or aspherical). They also present a method for deriving the particle size distributions from the measured data. Information about particle phases is desperately needed because mixed-phase clouds are still not well understood. Further, the full potential of the PHIPS-HALO probe has yet to be realized because both particle size distributions and particle phase distributions have not been routinely been made available in a short time period after the conduct of field campaigns. As the material in this paper works to overcome both of these shortcomings, it should be published as soon as possible and represents a good contribution to the refereed literature. The paper is well written and technically sound so does not require a lot of revisions in that respect. Nevertheless, I think there are a couple of aspects that should be better explained in the paper so that the limitations, as well as the strengths, of the PHIPS-HALO for providing size and habit information are well outlined.

> We thank the reviewer for this encouraging general comment. Below we have addressed the proposed revisions.

My major critique of the paper is that I think more information about the statistical representativeness of the data that are available from the PHIPS HALO probe should be included. The sample volume and/or sample area of the PHIPS-HALO probe should be explicitly stated. How does that compare against the sample volume/area from the commonly used. How does that affect the averaging time over which representative particle size distributions and particle phase distributions are available? For example, McFarquhar et al. (2007) calculate the required time that particle size distributions would need to be averaged over in order to obtain statistically significant particle size distributions that they defined to mean 100 particles in each size bin (so that there was a 10% uncertainty assuming the statistical uncertainty was proportional to the square

root of the number of counts in each size bin). Figure 9 defines the sensitive area of the PHIPS-HALO as ranging from 0.002 $cm2$ to about 0.01 $cm2$ depending on the size of the particle. Assuming a roughly 150 m/2 air speed, this would give sample volumes ranging from 30 to 150 $cm3$ per second, or about 0.15 liter/second. This would seem to be quite a bit less than that of the 2DS/2DC class of probes. This does not negate the benefits of the PHIPS-HALO probe, but rather would seem to suggest that the more detailed phase/shape information available from the PHIPS-HALO probe has synergy with the more frequent data available from the optical array type probes that are better suited for deriving the fine resolution structure of clouds. This point, including explicitly comparing the sampled areas/volumes and numbers of particles between probes should be explicitly shown in the paper, and also mentioned in the abstract.

> *We recognise the concern of the reviewer and have included more information about the statistical representativeness of the PHIPS probe. We would like to remind that PHIPS as a single-particle probe needs to have significantly smaller sampling volume compared to shadow imagers to ensure that only one particle occupies the sampling volume when measurements take place. However, the sampling area of PHIPS does not significantly differ from other single-particle probes. Of course, single-particle instruments have the disadvantage of a smaller sampling volume compared to shadow imagers, which will affect the statistical representativeness of the measurements. Therefore, we added a section (4.4) discussing exemplary sample volume values and required averaging times, including a comparison with the common shadow imagers. Also, statistical uncertainty bars were added to the PSD figures for the three case studies in section 5.*

Detailed Comments Page 1, line 6: evaluated would be a better word than validated.

> *The phrasing was adjusted accordingly.*

Page 2, line 13: "a large sampling statistics is required" reads awkwardly and should be rephrased.

> *The phrasing was adjusted accordingly.*

Page 3, line 15: the plural of aircraft is aircraft.

> *The phrasing was adjusted accordingly.*

Page 4, line 7: Suggest adding Um et al. 2011 ACP to the list of references as they considered scattering functions of several models of quasi-spherical ice crystals

> *Um et al., 2011 was added to the list of references.*

Page 4, Figure 2: Can you state what are the maximum dimensions of the two ice crystals that are considered in the figure?

> *The diameter of the two exemplary particles used in the figure are 119.6 $\mu m$ for the droplet (blue) and 165.8 $\mu m$ for the ice particle (red). This was added to the caption of the figure.*

Page 7, line 23: It should be noted and discussed why there are a lot of differences in the nature of the distributions between the observed and modeled particles in Figures 4 and 5. Does this suggest that there are some limitations in how well the theoretical models are representing the actual observed particles?

> *We agree with the reviewer, that a detailed comparison and discussion of the feature-parameter-distribution-plots (Fig. 4 and Fig. 5) is missing. The following discussion was added in section 3.3:*
> *"The plots show that the distribution of the four aforementioned feature parameters are clearly distinct for droplets and ice and thus represent features that can be used to discriminate droplets from ice. Further, it can be seen that these normalized occurrences ($f_i$) are normally distributed. The distributions of the four feature parameters based on the measurements (Fig. 6)*

*show a similar trend to the simulations (Fig. 5). The width of the distributions of feature parameters for measurements is much broader compared to the simulations. The main reason for this is the single-orientation of the measured crystals compared to the orientation-averaging that was used in the simulations. Orientation-averaging tends to smooth out features in the ASFs and thus cause more narrow feature parameters. It should be also noted that the theoretical computations are for idealised crystals, whereas it can be assumed that individual atmospheric ice crystals can have ASFs that deviate from idealized shapes. Nevertheless, the mean values of the distributions agree very well. The only exception to this is the mean value of the distribution of droplets for f1, which is shifted slightly to larger values compared to the simulations. This is to be expected because the "Mie-comparison-feature" $f_1$ is based on the relative difference between the measured and calculated ASF. This difference is much smaller for simulated particles as discussed in 3.1.1."*

Page 9, line 3. This may be a stylistic thing, but when I see 41.000 I think there are 41 particles. I think the authors mean 41,000. Unless this convention is demanded by ACP, I would use a comma rather than a period.

> *The phrasing was adjusted accordingly throughout the whole document.*

Page 9, line 3: Related to my major point above, I think it would be very interesting to compare the number of particles that were measured by the 2DC/2DS for the same periods during these field projects. That would help clarify information about the statistical representativeness of the data.

> *We agree, in general, on the importance of a comparison of the PHIPS and 2DC/2DS probes, including the discussion about sampling rates and statistical representativeness. This discussion takes place later in the text (see answer to the major comment above). However, at this particular location*

> *in the text, a comparison with the total number of measured particles by the 2DC/2DS would not be representative since we have restricted our calibration dataset to carefully selected particles with available stereo-images. Only a sub-set of triggered particles have a stereo-image since the imaging rate of the camera is restricted. Additionally, only images that show clearly distinctive particles are used, whereas images that show cut-off, out-of-focus or multiple particles, are excluded. Furthermore, the calibration of the phase discrimination algorithm is based on images of individual particles, hence, for this particular application, the sampling rate is not relevant.*

Page 11, Figure 6. Was any effort made to go back and look at the particles that were misidentified to determine why they were misidentified? I agree that 98% is outstanding (and better than classifications that are based on other probe data), but it would still be interesting to know why the discrepancy for just these few particles. Was there any chance that the manual identification of these particles was incorrect?

> *All misclassified particles (in total 289) were investigated individually and the manual classification was double-checked. An overview of examples of misclassified particles and (possible) reasons for their misclassification are given in the SI (S4).*

Page 13, Figure 8. Most probes that measure small particles have smaller sensitive areas for smaller particles than larger particles (e.g., see Figure 9 in this paper). But, as I understand it the vertical axis here is number of particles rather than some measure of concentration per bin or number distribution function. It would be informative to include another plot that shows the calculated number distribution function since that is a physically meaningful quantity, especially since the caption reads that this is a particle size distribution (PSD).

> *The PSD distribution in Fig. 8 is showing the comparison (or rather agreement) of the PSD based on the ASF and the images for a sub-sample of*

*particles with stereo-images. Therefore, a number concentration would not be representative for the cloud and we give the concentrations in arbitrary units.*

Page 18, Figure 11. The caption should specifically state the averaging period for which each of the plotted points corresponds to.

*Each point corresponds to an averaging period of 30 seconds. This was added in the caption accordingly.*

Page 22, Line 10. To make this study more accessible, it would be nice to have the codes used available on github or some other code repository.

*We agree with the reviewer. The current version of the code can be accessed via https:// doi.org/ 10.5281/ zenodo.4321316.*